# Learning to Abstain in the Presence of Uninformative Data

## Abstract

Learning and decision making in domains with naturally high noise-to-signal ratios – such as Finance or Public Health – can be challenging and yet extremely important. In this paper, we study a problem of learning on datasets in which a significant proportion of samples does not contain useful information. To analyze this setting, we introduce a noisy generative process with a clear distinction between uninformative/not learnable/purely random data and a structured/informative component. This dichotomy is present both during the training and in the inference phase. We propose a novel approach to learn under these conditions via a loss inspired by the selective learning theory. By minimizing the loss, our method is guaranteed to make a near-optimal decision by simultaneously distinguishing structured data from the non-learnable and making predictions, even in a highly imbalanced setting. We build upon the strength of our theoretical guarantees by describing an iterative algorithm, which jointly optimizes both a predictor and a selector, and evaluate its empirical performance under a variety of conditions.

## 1 Introduction

Despite the success of machine learning in computer vision (Deng et al., 2009; Krizhevsky et al., 2009; He et al., 2016a; Huang et al., 2017) and natural language processing (Vaswani et al., 2017; Devlin et al., 2018), the power of ML is yet to make significant impact in other areas. One major challenge is the inherently high noise-to-signal ratio in certain domains. For example, in Finance, while stock prices generally reflect the information about financial health of their companies, over the short term, their fluctuations most closely resemble random walks - which are naturally unpredictable - and are usually modeled as such (Tsay, 2005). In biomedical research, the underlying phenomena are often highly complex and are affected by unobservable factors. The outcome may appear highly random to the measurements (gene expression, medical histories), if the true causing factor is not included or is overwhelmed by others.

We are interested in dealing with datasets that may contain large fraction of noisy/uninformative/not learnable data in both training and testing stages. Direct application of standard supervised learning methods to such datasets is both challenging and unwarranted. At the training stage, the uninformative data can significantly bias the model or even completely overwhelm the true signal (Nettleton et al., 2010). Therefore, naïve forecasts in majority-uninformative datasets are doomed to be unreliable. Compared to other learning methods, deep neural networks are even more affected by the presence of noise, due to their strong memorization power (Zhang et al., 2017): they are likely to overfit the noise and make overly confident predictions where no real structure exists.

In this paper, we propose a novel method for learning on datasets where a significant portion of content is pure noise. Instead of forcing the classifier to make predictions for every sample, we learn to decide whether a datapoint is informative or not. If successful, the method abstains from making decisions where no structure exists, and predominantly learns from the remaining predictable data.

Our idea is inspired by the classic selective prediction problem (Chow, 1957), in which one learns to select a subset of data and only predict on that subset. However, the goal of selective prediction is very different from ours. A selective prediction method considers all data relevant. It pursues a balance between coverage (i.e. proportion of the data selected) and conditional accuracy on the selected data. In our problem, we assume that uninformative data is an integral part of the data generative process. No learning method, no matter how powerful, can be successful on such data. Our goal is to identify these uninformative samples as well as possible, and at the same time, to train a classifier by minimizing conditional risk on the remaining informative data.

Our method learns both a *predictor*, $f$, that classifies samples, and a *selector*, $g$, that selects learnable data for the predictor and rejects/abstains from the uninformative data. We are using $g$ to approximate the ground truth indicator function of structured/informative data, $g^*$. We assume that $g^*$ exists as a part of the data generation process, but it is never revealed to us, even during training. Instead of direct supervision, we therefore must rely on the predictor's mistakes to train the selector. To achieve this goal, we propose a novel *selector loss* enforcing that (1) the selected data best fits the predictor, and (2) the portion of the data where we abstain from forecasting, does not contain many correct predictions. This loss function is quite different from the loss in classic selective prediction, which penalizes all unselected data equally.

A major contribution of this paper is the derivation of theoretical guarantees for the empirical minimizer of our loss. We analyze the proposed selector loss function and provide sample complexity for learning a nearly optimal selector. We show that optimizing such loss can recover nearly all the structured/informative data in a PAC fashion (Valiant, 1984; Kearns et al., 1994; Valiant, 2013), i.e. one can approximate the ground truth selector function $g^*$ well with high probability, given sufficient samples. What may be surprising is that this guarantee holds even in a challenging setting where the uninformative data represents the majority of the training set.

This theoretical guarantee lets us expand to a more challenging and realistic setting. When the sample size is limited and the initially learned predictor is not sufficiently close to the ground truth, we extend our method to an iterative algorithm, in which we progressively optimize both the predictor and the selector. The selector is improved by optimizing our novel selector loss. Meanwhile, the predictor is improved by optimizing the empirical risk, reweighted based on the selector's output; uninformative or nearly-uninformative samples identified by the selector will be down-weighed. Experiments on real-world datasets demonstrate superiority of our method to existing baselines. Note that in this paper, we assume that each sample is either informative or not. Extending our method to a more general setting with continuous transitions between informative and uninformative is non-trivial and is left as future work.

## 1.1 RELATED WORK

**Learning with untrusted data** aims to recover the ground truth model from a partially corrupted dataset. Different noise models for untrusted data have been studied, including random label noise (Bylander, 1994; Natarajan et al., 2013; Han et al., 2018; Yu et al., 2019; Zheng et al., 2020; Zhang et al., 2020), bounded label noise (Massart & Nédélec, 2006; Awasthi et al., 2015; Diakonikolas et al., 2019; 2020) and adversarial noise (Kearns & Li, 1993; Kearns et al., 1994; Kalai et al., 2008; Klivans et al., 2009; Awasthi et al., 2017). In the most pessimistic setting, if the majority data is corrupted by arbitrary adversarial noise, even mean estimation may be impossible. This is known as List-Decodable problem (Balcan et al., 2008; Charikar et al., 2017; Diakonikolas et al., 2018), where the best one can do when the proportion of trusted data $\alpha$ is less than 0.5, is to return $\frac{1}{\alpha}$ many hypotheses of the mean, knowing one of them is promising. We, on the other hand, aim to produce a single accurate model even in a setting where the majority of the data is uninformative (the noise, of course, must be of a certain type, see next section). While above works assume the presence of noisy data only in the training stage, we study the case where noise is an integral part of the generative process and thus will appear during inference as well, where it must be detected and discarded once more.

**Selective learning** is an active research area. It extends the classic selective prediction problem and studies how to select a subset of data for different learning tasks. We can summarize existing methods into 4 categories: Monte Carlo sampling based methods (Gal & Ghahramani, 2016; Kendall & Gal, 2017; Pearce et al., 2020), margin based methods (Fumera & Roli, 2002; Bartlett & Wegkamp, 2008; Grandvalet et al., 2008; Wegkamp et al., 2011; Zhang et al., 2018), confidence based methods (Wiener & El-Yaniv, 2011; Geifman & El-Yaniv, 2017; Jiang et al., 2018) and customized selective loss (Cortes et al., 2016; Geifman & El-Yaniv, 2019; Liu et al., 2019). Notably, several works propose customized losses, and incorporate them into neural networks. In (Geifman & El-Yaniv, 2019), the network maintains an extra output neuron to indicate rejection of datapoints. Liu et al. (2019) use Gambler loss where a cost term is associated with each output neuron and a doubling-rate-like loss function is used to balance rejections and predictions. Cortes et al. (2016) perform data selection with an extra model and introduce a selective loss that helps maximize the coverage ratio, thus trading off a small fraction of data for a better precision.

Existing works on selective prediction are all motivated from the coverage perspective - i.e. one wants to make safe prediction to achieve higher precision while maintain a reasonable recall (El-Yaniv et al., 2010). Whereas our paper is the first to investigate the case where some (or even majority) of the data is uninformative, and thus must be discarded at prediction time. Unlike with the selective prediction, there is a latent ground truth indicator function of whether a datapoint should be selected or not. Our method is guaranteed to identify those uninformative samples.

## 2 PROBLEM FORMULATION

In this section, we describe the inherently-noisy data generation process that we aim to study. The model has three important features: 1) the uninformative portion of the data has labels generated by coin flipping; 2) the informative datapoints are labeled with a latent ground truth function; and 3) the uninformative data has a distinguishable support from the informative data. Formally:

**Definition 1** (Noisy Generative Process). *We define* Noisy Generative Process *by the following notation* $x \sim \mathcal{D}_\alpha$ *where*

$$\mathcal{D}_\alpha \equiv \begin{cases} x \sim \mathcal{D}_U & \text{with prob. } 1 - \alpha \quad \textbf{(Uninformative/Noisy Data)} \\ x \sim \mathcal{D}_I & \text{with prob. } \alpha \qquad \textbf{(Informative/Structured Data)}. \end{cases} \tag{1}$$

*Let* $\Omega_\mathcal{D} \subseteq \mathbb{R}^d$ *be the support of* $\mathcal{D}_\alpha$. *Suppose* $\{\Omega_U, \Omega_I\}$ *is a partition of* $\Omega_\mathcal{D}$, *the ground truth labeling function* $f^* : \mathcal{X} \to \{+1, -1\}$ *is in hypothesis class* $\mathcal{F}$ *and data is sampled according to:*

$$x \sim \mathcal{D}_\alpha; y \equiv \begin{cases} Bernoulli(0.5), & \text{if } x \in \Omega_U \\ f^*(x), & \text{if } x \in \Omega_I. \end{cases} \tag{2}$$

Note that the $f^*$ is defined on the whole domain, although only the part within $\Omega_I$ is relevant. $\alpha$ represents the fraction of informative or structured data in the population. For the rest of the paper, we abuse the notation and use $(x, y) \sim \mathcal{D}_\alpha$ to refer to samples generated from this process.

Next definition describes a separability condition between informative and uninformative data. It allows to distinguish noise from structure and enables us to approximate the ground truth selector via empirical minimization.

**Definition 2** ($\mathcal{H}$-Separable). *Given compact set* $\Omega$ *and its partition* $\{\Omega_U, \Omega_I\}$, $\{\Omega_U, \Omega_I\}$ *satisfies* $\mathcal{H}$-Separable *condition if there exists* $g^* \in \mathcal{H}$ *satisfying* $g^*(x) : \mathcal{X} \to \{+1, -1\}$:

$$g^*(x) \equiv \begin{cases} -1, & \text{if } x \in \Omega_U \\ 1, & \text{if } x \in \Omega_I. \end{cases} \tag{3}$$

One can view $g^*(\cdot)$ in definition 2 as the target selector we wish to recover. Having introduced the data generation process and the separability condition, we now describe our main assumption for the remainder of the paper.

**Assumption 1.** *Data* $S_n = \{x_i, y_i\}_{i=1}^N$ *is i.i.d generated according to the* Noisy Generative Process *(Definition 1), with* $f^* \in \mathcal{F}$, *support* $\Omega_{\mathcal{D}_\alpha} = \Omega_U \dot\cup \Omega_I$ *where* $\Omega_U, \Omega_I$ *are* $\mathcal{H}$-Separable.

Throughout this paper, we are interested in the following learning task:

**Problem 1** (Abstain from Uninformative Data). *Under Assumption 1 with enough i.i.d observations from* $\mathcal{D}_\alpha$: *1) given hypothesis class* $\mathcal{F}$, *we aim to learn the underlying ground truth classifier* $f^*(x)$; *and 2) given the hypothesis class* $\mathcal{H}$, *we aim to learn the selector* $g^*(x)$.

**Evaluation metrics.** We define metrics to evaluate the quality of both the prediction and the selection. For the prediction, we borrow the selective risk definition from selective learning (El-Yaniv et al., 2010; Geifman & El-Yaniv, 2019; 2017) using our own notation.

**Definition 3** (Selective Risk). *Given a predictor* $f \in \mathcal{F}$ *and a selector* $g \in \mathcal{H}$, *we define the selective risk as:* $CR(f, g) = \mathbb{E}_{(x,y) \sim \mathcal{D}_\alpha}[\mathbb{1}\{f(x) \neq y\}|g(x) \geq 0]$ *and its empirical version:* $CR_S(f, g) = \frac{\sum_{i=1}^n \mathbb{1}\{f(x_i) \neq y_i\}\mathbb{1}\{g(x_i) \geq 0\}}{\sum_{i=1}^n \mathbb{1}\{g(x_i) \geq 0\}}$.

Selective risk measures the average risk conditioned on instances that are picked by the selector $g(x)$. Note here when combined with the ground truth selector $g^*$, the selective risk of the ground truth

predictor $f^*$ goes to zero. Without a selector, however, $f^*$ has classical classification risk (which will be formally defined in Definition 5 later) of more than $\frac{1}{2}(1 - \alpha)$.

The metric used to evaluate the quality of a learned selector $g$ is its false positive/negative rate. When $g(x) \geq 0$ and $g^*(x) < 0$, the selector $g$ accepts an uninformative datapoint and thus commits a *false positive error*. When $g(x) < 0$ and $g^*(x) \geq 0$, $g$ rejects/abstains from an informative datapoint resulting in a *false negative error*.

**Definition 4** (Evaluation Metric for the Selector). *Given distribution $\mathcal{D}_\alpha$ as defined in Defnition 1 and $g^*$ as target selector, we denote the false positive and false negative of a selector $g$ as*

$$\text{False Positive: } \mathbb{P}[g(x) \geq 0 | g^*(x) < 0]; \text{ False Negative: } \mathbb{P}[g(x) < 0 | g^*(x) \geq 0] \tag{4}$$

## 3 OUR METHOD

In this section, we present our approach for learning and abstaining in the presence of uninformative data (Problem 1). The main challenge is that the latent informative/uninformative status of a datapoint is unknown. Our main idea is to introduce a novel *selector loss* function that trains a selector based on the performance of the best predictor (Section 3.1). In Section 3.2, we present our main theoretical result. We show that by jointly finding a predictor minimizing the classification risk and finding a selector minimizing the proposed selector loss, we can solve Problem 1 with controlled selective risk and selector error rates. During the analysis, we assume an oracle is given for the empirical minimizer. Approximating such empirical minimizer is a different topic beyond the scope of this paper (Bartlett et al., 2006; Yuan & Wegkamp, 2010). Inspired by the theoretical results, in Section 3.3, we propose a heuristic algorithm that iteratively optimizes the predictor and the selector.

Throughout the analysis, we focus on the 0-1 classification loss to optimize the predictor. This follows the standard for analyzing generalization performance and sample complexity. Formally:

**Definition 5** (Classification Risk). *We denote the classification risk of a model $f$ as*

$$R(f) \equiv \mathbb{E}_{(x,y) \sim \mathcal{D}_\alpha}[\mathbb{1}\{f(x) \neq y\}] \tag{5}$$

*and the empirical classification risk: $R_{S_n}(f) = \frac{1}{n}\sum_{i=1}^n \mathbb{1}\{f(x_i) \neq y_i\}$. We define $f^*_{S_n}$ to be the empirical minimizer of $R_{S_n}(f)$: $f^*_{S_n} = \arg\min_{f \in \mathcal{F}} R_{S_n}(f)$.*

### 3.1 SELECTOR LOSS

To learn a selector without direct supervision, we have to leverage the performance of a given predictor, $f$. We propose the *selector loss*, which reweigh the 0-1 loss based on the prediction of $f$. This loss penalizes when (1) the predictor makes a correct prediction on a datapoint that the selector considers uninformative and abstains from, or (2) the predictor makes an incorrect prediction on a datapoint that the selector considers informative.

**Definition 6** (Selector Loss). *Given $f \in \mathcal{F}$ and its selector $g \in \mathcal{H}$, we define the following weighted $0 - 1$ type risk w.r.t $g(\cdot)$ as selector risk:*

$$W(g; f, \theta) \equiv \mathbb{E}_{(x,y) \sim \mathcal{D}_\alpha}\left[\frac{\theta}{\alpha}\mathbb{1}\{f(x) \neq y\}\mathbb{1}\{g(x) > 0\} + \frac{1-\theta}{1-\alpha}\mathbb{1}\{f(x) = y\}\mathbb{1}\{g(x) \leq 0\}\right].$$

*and its empirical version:*

$$W_{S_n}(g; f, \theta) = \frac{1}{n}\sum_{i=1}^n \frac{\theta}{\alpha}\mathbb{1}\{f(x_i) \neq y_i\}\mathbb{1}\{g(x_i) > 0\} + \frac{1-\theta}{1-\alpha}\mathbb{1}\{f(x_i) = y_i\}\mathbb{1}\{g(x_i) \leq 0\} \tag{6}$$

Intuitively speaking, the loss will drive the selector to partition the domain into informative and uninformative regions. Within the informative region, the predictor is supposed to fit the data well, and should be more accurate. Meanwhile, within the uninformative region, the label is random and the predictor is supposed to be more prone to error. Another view of the loss is that we are learning to fit the selector $g(x)$ to a *pseudo-informative label* given by the predictor, $\mathbb{1}\{f(x) = y\}$. Since such label is $f$-dependent, the quality of $f$ is crucial for successfully learning $g$. In the theoretical analysis, we leverage the fact that predictor $f$ is close enough to the ground truth predictor $f^*$.

Note that there are two types of errors penalized in the selector loss: an incorrect prediction on an informative datapoint, $(f(x) \neq y) \wedge (g(x) > 0)$, and a correct prediction on an uninformative datapoint, $(f(x) = y) \wedge (g(x) \leq 0)$. We use $\beta_I = \frac{\theta}{\alpha}$ and $\beta_U = \frac{1-\theta}{1-\alpha}$ to weigh these two types of error in the loss. Both $\beta_I$ and $\beta_U$ are controlled by a parameter $\theta$ and the latent informative data proportion $\alpha$. We show that for a wide range of $\theta$, the accuracy of the selector is guaranteed, as long as $\alpha$ is bounded away from 1, i.e., when abstention is necessary. In practice, we can adjust $\beta_I$ and $\beta_U$ freely without knowing exact value of $\alpha$, as empirical performance remains stable with regard to these choices.

**Learning a selector with the novel loss.** In an ideal setting, to learn a selector, we first estimate the predictor by minimizing the classification risk, $f_{S_n}^* = \arg\min_{f \in \mathcal{F}} R_{S_n}(f)$. Next, we estimate the selector by minimizing the selector loss, conditioned on the estimated predictor, $f_{S_n}^*$, $g_{S_n}^* = \arg\min_{g \in \mathcal{H}} W_{S_n}(g, f_{S_n}^*, \theta)$. In Figure 1, we show an example of using the empirical minimization strategy with logistic regression and with 0-1 loss replaced by cross-entropy loss. In this case, the losses are all convex and the empirical minimizers $f_{S_n}^*$ and $g_{S_n}^*$ can be computed exactly. Our analysis in Section 3.2 will show that the estimated predictor $f_{S_n}^*$ and selector $g_{S_n}^*$ has bounded selective risk (as in Definition 3), as well as bounded false positive and false negative selector errors (as in Definition 4).

In practice, however, empirical minimization is not always possible, as optimization for complex models (e.g., DNNs) and non-convex losses remains open. We therefore propose a heuristic algorithm in the spirit of our theoretical results - it jointly learns $f$ and $g$ by minimizing the selector loss and a reweighed classification risk iteratively (see Section 3.3).

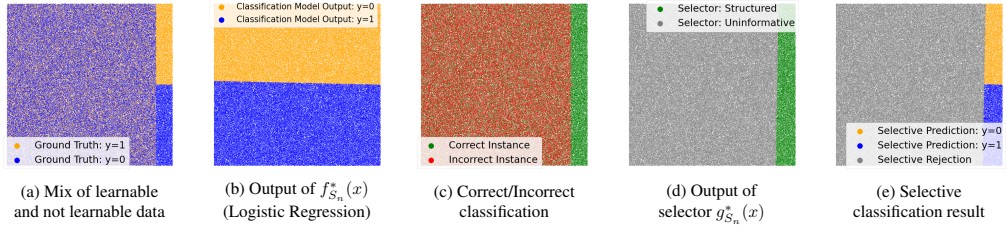

(a) Mix of learnable and not learnable data

(b) Output of $f_{S_n}^*(x)$ (Logistic Regression)

(c) Correct/Incorrect classification

(d) Output of selector $g_{S_n}^*(x)$

(e) Selective classification result

Figure 1: Illustration of the learning strategy in an ideal setting, using synthetic (uniformly distributed noise) data. We replace the 0-1 loss with cross-entropy loss and train logistic regression models for both $f$ and $g$. (a) shows that the informative and the uninformative supports are separated. (b) demonstrates that the model derived with majority-uninformative data has reasonable performance on the informative portion. (c) shows that the predictor has high accuracy in the informative region, but low accuracy in the uninformative region. In (d) and (e), the selector trained with $f_{S_n}^*$ successfully recovers informative support thus resulting in low selective risk.

### 3.2 THEORETICAL ANALYSIS

In this section we report our theoretical results. The main result can be summarized in the following (informal) statement.

**Main Result (Informal)** With sufficient data, the estimated predictor $f_{S_n}^* = \arg\min_{f \in \mathcal{F}} R_{S_n}(f)$ and selector $g_{S_n}^* = \arg\min_{g \in \mathcal{H}} W_{S_n}(g, f_{S_n}^*, \theta)$ are sufficiently close to the targets $f^*$ and $g^*$ with high probability.

The proof requires the classic growth function definition (Vapnik & Chervonenkis, 2015).

**Definition 7** (Growth Function). *Let $\mathcal{H}$ be the hypothesis class of function $f$ and $\mathcal{F}_{x_1,...,x_n} = \{f(x_1),...,f(x_n) : f \in \mathcal{F}\} \subseteq \{+1,-1\}^n$. The growth function is defined to be the maximum number of ways in which $n$ points can be classified by the function class: $\mathcal{G}_{\mathcal{F}}(n) = \sup_{x_1,...,x_n} |\mathcal{F}_{x_1,...,x_n}|$.*

**Roadmap of the proof**: Theorem 1 is a standard PAC Learning analysis where we show $f_{S_n}^*$ is sufficiently close to the ground truth function $f^*$. Note that our analysis does not imply efficient PAC learning since optimizing binary loss is NP-hard. In Theorem 2, we bound the statistical error of the selector loss. In Theorem 3, we leverage the fact that informative and uninformative portions of the data are distinguishable, as well as the fact that $f_{S_n}^*$ has low risk on informative data. This lets us bound the selector loss of $g_{S_n}^*$ given by $W(g_{S_n}^*, f_{S_n}^*, \theta)$. By carefully balancing the weight parameter

$\theta$ of selector loss, we derive the false positive and false negative guarantees in Theorem 3. Theorem 4 is a formal version of the main result, which applies Sauer's Lemma (Sauer, 1972) to bound the growth function of hypothesis classes in Theorem 3. The toolkit we use is driven by the concept of a VC-dimension in (Vapnik & Chervonenkis, 2015; Blum et al., 2016; Mohri et al., 2018). We also provide a lower bound construction in the Appendix to show the (near) tightness of our analysis (up to a logarithmic factor). We present all the results below, detailed proofs are in the Appendix.

We first prove a theorem about sample complexity of the learning hypothesis optimized over a given dataset, referred to as $f_{S_n}^*$. We show that it approximates the ground truth function $f^*$ well.

**Theorem 1.** *Under Assumption 1, given hypothesis class $\mathcal{F}$, a set of samples $S_n = \{(x_1, y_1), ..., (x_n, y_n)\}$ drawn i.i.d from our Noisy Generative Process and $f_{S_n}^* = \arg\min_{f \in \mathcal{F}} R_S(f)$, suppose $n \geq \frac{32 \left[ \log(4|\mathcal{G}_{\mathcal{F}}(2n)|) + \log(\frac{1}{\delta}) \right]}{\epsilon^2 \alpha^2}$, then with probability at least $1 - 2\delta$, $R(f_{S_n}^*) \leq \frac{1}{2}(1 - \alpha) + 2\epsilon\alpha$. Furthermore, $\mathbb{P}_{x \sim \mathcal{D}_I}[f^*(x) \neq f_{S_n}^*(x)] \leq 2\epsilon$.*

Next theorem states the sample requirements in order to achieve a small gap between the empirical selector loss and the true selector loss. The proof takes a union bound over all possible pairs of hypothesis $(g, f) \in \mathcal{H} \times \mathcal{F}$, which explains the term $|\mathcal{G}_{\mathcal{H}}(2n)||\mathcal{G}_{\mathcal{F}}(2n)|$ in the bound for $n$, the minimum number of sample points.

**Theorem 2.** *Given hypothesis class $\mathcal{H}, \mathcal{F}$, a set of samples $S_n = \{(x_1, y_1), ..., (x_n, y_n)\}$ drawn i.i.d. from the Noisy Generative Process and $f_{S_n}^* = \arg\min_{f \in \mathcal{F}} R_S(f), g_{S_n}^* = \arg\min_{g \in \mathcal{H}} W_{S_n}(g, f_{S_n}^*, \theta)$, if $\theta \in (\alpha, 1), \alpha \in (0, 1)$ and $n \geq \frac{32\theta^2 \left[ log(4|\mathcal{G}_{\mathcal{H}}(2n)||\mathcal{G}_{\mathcal{F}}(2n)|) + \log(\frac{1}{\delta}) \right]}{\epsilon^2 \alpha^4}$, with probability at least $1 - \delta$,*

$$|W(g_{S_n}^*, f_{S_n}^*, \theta) - W_{S_n}(g_{S_n}^*, f_{S_n}^*, \theta)| \leq \epsilon\alpha.$$

Theorem 3 combines the results in Theorems 1 and 2. It shows that, by carefully choosing weight parameter $\theta$, minimizing the selector loss with sufficiently large amount of data ensures a low false positive/negative rate of the selector with high probability. In particular, the analysis points to a trade off between false positive and negative rates faced by the selector in terms of $\theta$. We characterize the relationship between $\theta$ and $\alpha$ using variable $c = \frac{\theta}{2\alpha}$ in the theorem and prove that for a wide range of $c$ the false negative rate (FNR) and false positive rate (FPR) are controlled. Intuitively, a larger value of $\theta$ in $W_{S_n}(g, f_{S_n}^*, \theta)$ implies a higher penalty on the false positive error.

**Theorem 3.** *Under Assumption 1, given a set of samples $S_n = \{(x_1, y_1), ..., (x_n, y_n)\}$ drawn i.i.d. from the Noisy Generative Process and $f_{S_n}^* = \arg\min_{f \in \mathcal{F}} R_S(f), g_{S_n}^* = \arg\min_{g \in \mathcal{H}} W_{S_n}(g, f_{S_n}^*, \theta)$, if $\theta \in (\alpha, 1), \alpha \in (0, 1)$ and $n \geq \frac{32\theta^2 \left[ log(4|\mathcal{G}_{\mathcal{H}}(2n)||\mathcal{G}_{\mathcal{F}}(2n)|) + \log(\frac{5}{\delta}) \right]}{\epsilon^2 \alpha^2}$, if $\theta = 2c\alpha$ with $\frac{1}{2} < c < \frac{1+\alpha}{4\alpha}$, we have with probability at least $1 - \delta$:*

$$\mathbb{P}[g_{S_n}^*(x) \geq 0 | g^*(x) \leq 0] \leq \frac{12c\alpha\epsilon}{2c - 1} \quad \textit{[False Positive]}$$
$$\mathbb{P}[g_{S_n}^*(x) < 0 | g^*(x) \geq 0] \leq 20c\epsilon \quad \textit{[False Negative]} \tag{7}$$

Next theorem is a formal statement of the main result. It is worth mentioning that the theorem assumes that $\alpha$ is bounded away from 1, due to the fact that when almost all data is informative, there is no need for abstention.

**Theorem 4.** *Under Assumption 1, given set of samples $S_n = \{(x_1, y_1), ..., (x_n, y_n)\}$ drawn i.i.d. from the Noisy Generative Process, hypothesis class $\mathcal{H}$ with VC-dimension $d_{vc}(\mathcal{H})$, $\mathcal{F}$ with VC-dimension $d_{vc}(\mathcal{F})$ and $f_{S_n}^* = \arg\min_{f \in \mathcal{F}} R_S(f), g_{S_n}^* = \arg\min_{g \in \mathcal{H}} W_{S_n}\left(g, f_{S_n}^*, \frac{73\alpha}{72} + \frac{\alpha^2}{72}\right)$, if $\alpha \in (0, 0.9)$ and $n \geq \frac{128 \left[ (d_{vc}(\mathcal{H}) + d_{vc}(\mathcal{F})) \log(\frac{1}{\varepsilon}) + \log(\frac{16}{\delta}) \right]}{\epsilon^2 \alpha^2}$. We have with probability at least $1 - \delta$:*

*I)*
$$\mathbb{P}[g_{S_n}^*(x) \geq 0 | g^*(x) \leq 0] = O(\epsilon) \quad \textit{[False Positive]}$$
$$\mathbb{P}[g_{S_n}^*(x) < 0 | g^*(x) \geq 0] \leq O(\epsilon) \quad \textit{[False Negative]} \tag{8}$$

*II)*
$$\mathbb{P}_{x \sim \mathcal{D}_\alpha}[f^*(x) \neq f_{S_n}^*(x) | g_{S_n}^*(x) \geq 0] = O(\epsilon)$$

**Remark 1.** *We provide a lower bound construction in the appendix to show that our bound in Theorem 4 is tight up to a logarithmic factor. Note that our lower bound is for **empirical minimizers** of $R(f)$ and $W(g, f, \theta)$, therefore the tightness result is not an information theoretic bound for this family of problems. It remains an open question about whether or not strategies other than the empirical minimization, as in (Hanneke, 2016; Simon, 2015; El-Yaniv et al., 2010), can improve the sample complexity to a rate of $O\left(\frac{1}{\alpha\varepsilon}\right)$. Note that the informative portion of the data (drawn with probability $\alpha$) by itself only requires $O(\frac{1}{\varepsilon})$ samples for an $\epsilon$-excessive statistical error. It turns out that our analysis for empirical minimizer is nearly tight, and that the number of samples $O\left(\frac{1}{\alpha^2\varepsilon^2}\right)$ is indeed necessary because the presence of noisy data makes our problem harder.*

**Remark 2.** *Theorem 4 says that by minimizing the empirical version of the loss from Definitions 5 and 6, one can achieve the goal of simultaneously selecting and learning with noisy data - i.e. the classification model $f^*_{S_n}$ has sufficiently low selective risk, and the selector $g^*_{S_n}$ can distinguish informative data from non-informative. The analysis of the selector loss (Theorem 3) relies on the quality of the classifier $f^*_{S_n}$. But since we know that $g^*_{S_n}$ is able to abstain from uninformative data, we can retrain $f^*_{S_n}$ on the informative portion only, therefore guaranteeing its accuracy. Such circular logic naturally leads to a practical iterative algorithm that we present in the next section.*

## 3.3 A Heuristic Algorithm in Practice

In Section 3.2, we analyzed an empirical minimizer of a conceptual 0-1 loss for the classifier $f$ and the selector $g$. Binary loss, however, is impractical from the computational standpoint. In practice, we use cross-entropy loss instead and require that both $f$ and $g$ have continuous-valued output, ranging between 0 and 1. We also relax the requirement for empirical minimization oracles, allowing the practical algorithm to jointly optimize the predictor and the selector in an iterative manner. At each iteration, we update the predictor using the informative data selected by the selector, and then update the selector based on the predictor's output. See Algorithm 1 for the pseudo-code. A pictorial example of Algorithm 1's performance can be found in Figure 2 in the Appendix.

During the joint optimization process, the predictor is counting on the selector to show it only the informative data. However, the initial selector is not trustworthy. To update the predictor $f$, we turn to a so-called soft abstention scheme: use a weight vector $\gamma$ that progressively down-weighs samples abstained by $g$, in the spirit of multiplicative weights algorithms (Cesa-Bianchi & Lugosi, 2006; Arora et al., 2012). Specifically, we increase the weight of $i$-th sample $\gamma_i$ if the selector accepts $x_i$: $\gamma_i = \gamma_i(1 + \eta \cdot \mathbb{1}\{g(x_i) > 0.5\})$ and then normalize so that $\sum_{i=1}^n \gamma_i = 1$. We call this a soft abstention approach because the algorithm decreases the weight of uninformative data gradually.

In the heuristic algorithm, we use cross-entropy loss to fit the selector to pseudo-informative labels output by the predictor. We also use a single hyper-parameter $\beta = \beta_I/\beta_U = \frac{\theta(1-\alpha)}{\alpha(1-\theta)}$ to control the ratio of the two types of errors. As suggested by Theorem 3, small selector FPR and small selector FNR can be achieved by a $\beta \in \left(1, 1 + \frac{1}{\alpha}\right)$. For a scenario when abstention is necessary, i.e. when the noisy data ratio is in a reasonable range, we can choose our hyper-parameter from a wide interval.

---

**Algorithm 1** `Iterative Soft Abstain`

---

**Require:** Data set $S_n = \{(x_1, y_1), ..., (x_n, y_n))\}$, weight parameter:$\beta$, random initial $f^0$ and $g^0$, initial sample weights $\gamma_i^0 = \frac{1}{n}, \forall i \in [n]$, meta learning rate $\eta$, number of iterations $T$

1: **for** $t \leftarrow 1, \cdots, T$ **do**
2:    Optimize loss to update predictor $f^t$:$\sum_{i=1}^n \gamma_i^t \{y_i \log(f(x_i)) + (1 - y_i) \log(1 - f(x_i))\}$
3:    Approximate the 'pseudo-informative label' : $z_i^t = \mathbb{1}\{\mathbb{1}\{f^t(x_i) > 0.5\} = y_i\}$
4:    Optimize loss to update selector $g^t$: $\sum_{i=1}^n \{z_i^t \log(g(x_i)) + \beta(1 - z_i^t) \log(1 - g(x_i))\}$
5:    Update sample weights using $g^t$: $\gamma_i^{t+1} = \frac{\gamma_i^t(1 + \eta\mathbb{1}\{g^t(x_i) > 0.5\})}{\sum_{j=1}^n \gamma_j^t(1 + \eta\mathbb{1}\{g^t(x_j) > 0.5\})}$.
6: **end for**
7: Output $f^T, g^T$

---

## 4 Experiments

**Datasets.** We test the efficacy of our heuristic algorithm (Algorithm 1), via semi-synthetic experiments. We synthesize the noisy data by mixing datasets from two different domains and uniformly shuffling all labels in one of them. Our first synthetic noisy dataset is composed of MNIST (LeCun

et al., 1998) and Fashion-MNIST (Xiao et al., 2017). We uniformly shuffle the labels of MNIST images and keep original labels of Fashion-MNIST. As a result, images from MNIST are uninformative while images from Fashion-MNIST are considered informative. The fact that one dataset has hand-written digits and the other one contains pictures of clothes ensures distinct supports. We mix these two datasets in different proportions to proxy for our Noisy Generative Process.

We also perform experiments on another synthetic dataset, where noisy and informative portions come from similar domains. To achieve this, we use SVHN (Netzer et al., 2011). We uniformly at random flip labels of classes 5-9 into classes 0-4 and mix them with intact data from classes 0-4. In this case, we consider the data from classes 5-9 uninformative. In this mixed dataset, informative and uninformative data have similar domains, which makes it more challenging to correctly identify informative data (and harder to argue separate support).

**Baselines.** We compare our method to two of the most recently proposed selective learning algorithms. (1) **SelectiveNet** (Geifman & El-Yaniv, 2019), which integrates an extra neuron as a data selector in the output layer and also introduces a loss term to control the coverage ratio; (2) **DeepGambler** (Liu et al., 2019), which also maintains an extra neuron for abstention and uses a doubling-rate-like loss term (i.e., gambler loss) to train the model. (3) We also create a third baseline that selects data using model prediction confidence, which we refer to as **Confidence**. The hypothesis is that informative data should have higher prediction confidence compared to uninformative data.

**Evaluation Metrics.** We use three criteria to jointly evaluate a selective learning outcome. (1) Selective risk (SR). Selective risk is the empirical risk measured over the datapoints selected by the algorithm. This is a metric that is also adopted in (Geifman & El-Yaniv, 2019; Liu et al., 2019). (2) Precision. Precision is the proportion of true informative datapoints among all the data picked out by the selector. (3) Recall. Recall is the proportion of true informative samples picked out by the selector out of all the informative samples in the dataset. SR evaluates the quality of the classifier, Precision and Recall are the standard ML metrics of the selector. An ideal selective learning algorithm should have low SR, high precision and high recall.

**Experiment setting.** We assume that the ratio of informative data, $\alpha$, is unknown to all methods. This is necessary in practice; the ratio and strength of noise are not known in most real world scenarios. For SelectiveNet and DeepGambler, such ratio is a required input. To run these baselines, we first run the original backbone for 60 epochs, and then estimate $\alpha$ using backbone's training accuracy. Assume that the backbone fits all of the informative data perfectly and also makes some correct guesses on noisy data with probability $\frac{1}{\text{num of classes}}$, then the frequency estimation of $\alpha$ is $\hat{\alpha} = \frac{\text{num of classes} * \text{train acc} - 1}{\text{num of classes} - 1}$ , which is the value we give to baselines. For more details about the experimental design, please refer to the Appendix (section D).

**Results and discussion.** Table 1 presents the result where we mix the entire uninformative dataset (shuffled MNIST) with different proportions of the informative dataset (Fashion-MNIST). Our algorithm outperforms all the baselines in this setting. If we use partial dataset by decreasing the number of noisy datapoints and repeating previous experiments with the same uninformative-to-informative data ratios, our algorithm wins by an even larger margin (Table 2).

We can see that both SelectiveNet and DeepGambler fall apart in the setting where informative data are sparse. Neither of them achieve a risk below 40% when we mix in only 25% of informative data. The reason is that both of these two methods are only designed to achieve predetermined coverage and cannot dynamically adapt to settings with unknown informative-to-uninformative data ratio. Our algorithm thus gains advantage by learning the underlying uninformative data labeling function in a data-driven manner.

Our algorithm also outperforms all the baselines on SVHN dataset where noisy data comes from a similar domain to the informative data. Each experiment is repeated 5 times. Mean and standard deviation are presented in Table 3 and Table 4.

We provide more experimental results in the Appendix (section D). We test the performance of each method in the case where $\alpha$ is given and in the case where datasets contain minority noisy data.

Table 1: (MNIST - Full Dataset - Unknown $\alpha$). Results on a synthetic dataset consisting of uninformative MNIST and informative Fashion-MNIST using the entirety of shuffled MNIST to proxy for noise.

| Uninformative Data Num. | Informative Data Num. | Criterion | Confidence | SelectiveNet | DeepGambler | Ours |
|---|---|---|---|---|---|---|
| 60000 | 15000 | SR(%) | 59.32±0.83 | 45.44±0.68 | 45.37±1.29 | **10.35±0.31** |
| | | Precision | 0.46±0.79 | 0.61 ± 0.01 | 0.61±0.01 | 1.00 ± 0.00 |
| | | Recall | 0.94±0.07 | 1.00 ± 0.00 | 1.00±0.00 | 0.85 ± 0.01 |
| 60000 | 30000 | SR(%) | 39.23±0.47 | 28.15±0.63 | 27.78±0.67 | **12.03±1.06** |
| | | Precision | 0.68±0.56 | 0.80 ±0.01 | 0.80±0.00 | 1.00 ± 0.00 |
| | | Recall | 0.98±0.00 | 1.00 ± 0.00 | 1.00±0.00 | 0.92 ± 0.04 |
| 60000 | 45000 | SR(%) | 28.92±0.49 | 20.22±0.58 | 19.09±0.41 | **12.58±2.00** |
| | | Precision | 0.79±0.29 | 0.88 ±0.86 | 0.89±0.00 | 1.00 ± 0.00 |
| | | Recall | 0.99±0.00 | 1.00 ± 0.00 | 1.00 ± 0.00 | 0.97 ± 0.03 |
| 60000 | 60000 | SR(%) | 21.88±0.39 | 15.38±0.32 | 14.20±0.62 | **11.61±0.79** |
| | | Precision | 0.86±0.46 | 0.92±0.22 | 0.94±0.00 | 1.00 ± 0.00 |
| | | Recall | 1.00±0.00 | 1.00 ± 0.00 | 1.00 ± 0.00 | 0.97 ± 0.01 |

Table 2: (MNIST - Partial Dataset - Unknown $\alpha$) Results on a synthetic dataset consisting of uninformative MNIST and informative Fashion-MNIST data using 25% of shuffled MNIST for noise.

| Uninformative Data Num. | Informative Data Num. | Criterion | Confidence | SelectiveNet | DeepGambler | Ours |
|---|---|---|---|---|---|---|
| 15000 | 3750 | SR(%) | 79.87 ± 0.40 | 74.03 ± 1.38 | 73.22±0.51 | **20.24 ± 9.25** |
| | | Precision | 0.22 ± 0.00 | 0.29 ± 0.02 | 0.30 ± 0.01 | 1.00 ± 0.00 |
| | | Recall | 0.85 ± 0.00 | 1.00 ± 0.00 | 1.00 ± 0.00 | 0.88 ± 0.07 |
| 15000 | 7500 | SR(%) | 65.83 ± 0.22 | 57.76 ± 1.97 | 58.13±0.68 | **13.71±0.24** |
| | | Precision | 0.38 ± 0.00 | 0.48 ± 0.02 | 0.48 ± 0.01 | 1.00 ± 0.00 |
| | | Recall | 0.92 ± 0.01 | 1.00 ± 0.01 | 1.00 ± 0.00 | 0.83 ± 0.01 |
| 15000 | 11250 | SR(%) | 55.99 ± 0.49 | 47.39 ± 1.85 | 46.96±0.50 | **13.06±1.86** |
| | | Precision | 0.49 ± 0.01 | 0.60 ± 0.02 | 0.60 ± 0.01 | 0.99 ± 0.00 |
| | | Recall | 0.94 ± 0.01 | 1.00 ± 0.00 | 1.00 ± 0.00 | 0.84 ± 0.02 |
| 15000 | 15000 | SR(%) | 48.51 ± 0.25 | 39.23 ± 0.90 | 39.85±0.36 | **19.10±4.50** |
| | | Precision | 0.57 ± 0.00 | 0.68 ± 0.00 | 0.68 ± 0.00 | 0.99 ± 0.01 |
| | | Recall | 0.95 ± 0.00 | 1.00 ± 0.00 | 1.00 ± 0.00 | 0.91 ± 0.04 |

Table 3: (SVHN - Full Dataset - Unknonw $\alpha$) Results on a synthetic dataset consisting of uninformative SVHN and informative SVHN using all of shuffled classes to proxy for noisy data.

| Uninformative Data Num. | Informative Data Num. | Criterion | Confidence | SelectiveNet | DeepGambler | Ours |
|---|---|---|---|---|---|---|
| 33800 | 9200 | SR(%) | 64.91 ± 0.89 | 48.61 ± 23.37 | 18.24 ± 2.00 | **7.03±0.01** |
| | | Precision | 0.53 ± 0.01 | 0.52 ± 0.22 | 0.80 ± 0.02 | 0.93 ± 0.01 |
| | | Recall | 0.98 ± 0.00 | 0.89 ± 0.10 | 0.96 ± 0.01 | 0.85 ± 0.01 |
| 33800 | 18300 | SR(%) | 47.64 ± 0.89 | 28.15 ± 22.74 | 12.03 ± 0.99 | **5.41±0.97** |
| | | Precision | 0.70 ± 0.01 | 0.75 ± 0.18 | 0.88 ± 0.02 | 0.95 ± 0.01 |
| | | Recall | 0.99 ± 0.00 | 0.96 ± 0.10 | 0.98 ± 0.01 | 0.88 ± 0.01 |
| 33800 | 26200 | SR(%) | 36.53 ± 1.57 | 12.96 ± 1.70 | 8.65 ± 0.44 | **4.05±0.67** |
| | | Precision | 0.79 ± 0.01 | 0.88 ± 0.01 | 0.91 ± 0.01 | 0.97 ± 0.01 |
| | | Recall | 0.99 ± 0.00 | 0.98 ± 0.00 | 0.98 ± 0.00 | 0.89 ± 0.01 |
| 33800 | 28400 | SR(%) | 34.48 ± 1.15 | 12.10 ± 1.21 | 8.31 ± 0.64 | **4.43±0.86** |
| | | Precision | 0.80 ± 0.01 | 0.89 ± 0.01 | 0.92 ± 0.01 | 0.96 ± 0.01 |
| | | Recall | 0.99 ± 0.00 | 0.98 ± 0.00 | 0.98 ± 0.00 | 0.89 ± 0.01 |

Table 4: (SVHN - Partial Dataset - Unknown $\alpha$) Results on a synthetic dataset consisting of uninformative SVHN and Informative SVHN using 25% of shuffled classes to proxy for noisy data.

| Uninformative Data Num. | Informative Data Num. | Criterion | Confidence | SelectiveNet | DeepGambler | Ours |
|---|---|---|---|---|---|---|
| 9200 | 2285 | SR(%) | 74.47 ± 4.15 | 64.68 ± 20.32 | 34.48 ± 13.46 | **14.36 ± 0.08** |
| | | Precision | 0.41 ± 0.05 | 0.37 ± 0.22 | 0.68 ± 0.13 | 0.87 ± 0.07 |
| | | Recall | 0.94 ± 0.01 | 0.85 ± 0.23 | 0.83 ± 0.24 | 0.80 ± 0.02 |
| 9200 | 4600 | SR(%) | 61.44 ± 0.64 | 48.42 ± 23.42 | 26.42 ± 1.68 | **7.46 ± 0.87** |
| | | Precision | 0.57 ± 0.00 | 0.58 ± 0.17 | 0.77 ± 0.02 | 0.94 ± 0.07 |
| | | Recall | 0.95 ± 0.01 | 0.97 ± 0.03 | 0.96 ± 0.01 | 0.85 ± 0.01 |
| 9200 | 6900 | SR(%) | 52.41 ± 0.71 | 25.64 ± 3.01 | 20.83 ± 2.06 | **7.25±0.77** |
| | | Precision | 0.65 ± 0.00 | 0.76 ± 0.03 | 0.82 ± 0.02 | 0.94 ± 0.01 |
| | | Recall | 0.96 ± 0.00 | 0.96 ± 0.03 | 0.97 ± 0.01 | 0.87 ± 0.01 |
| 9200 | 9200 | SR(%) | 49.45 ± 0.45 | 24.69 ± 3.08 | 18.81 ± 1.47 | **7.22±0.54** |
| | | Precision | 0.67 ± 0.00 | 0.77 ± 0.03 | 0.83 ± 0.01 | 0.94 ± 0.01 |
| | | Recall | 0.96 ± 0.00 | 0.98 ± 0.00 | 0.98 ± 0.00 | 0.88 ± 0.01 |

## 5 CONCLUSION

In this work, we propose a natural noisy generative process to model a problem where a significant proportion of datapoints are purely random. We solve it by learning both a predictor for classification and a selector to abstain from the uninformative data. We propose a novel selector loss to learn selector with theoretical guarantees. Based on the selector loss, we design a heuristic algorithm that jointly learns the predictor and selector. Our empirical study shows promising results of our method.

## 6    REPRODUCIBILITY STATEMENT

We describe a synthetic data generation procedure, evaluation metrics and experiment setting in section 4. For the convenience of the reader to reproduce the experiment, we also summarize the setting and give implementation details in section D.3. The source code as well as parameters to reproduce the experimental results will be made available together with the publication of the paper.

## 7    ETHICS STATEMENT

This paper focuses on a theoretical discussion about learning from data that contains different portion of non-informative samples. Our experiments only use publicly available datasets. Our discussion, analysis, or data shouldn't raise any ethics-related issues. The learning method proposed in this paper, however, can be potentially used in applications with fairness and privacy concerns. It our future efforts in this area, we aim to address and resolve possible negative impact.

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

## A   ILLUSTRATIVE EXAMPLE FOR ALGORITHM 1

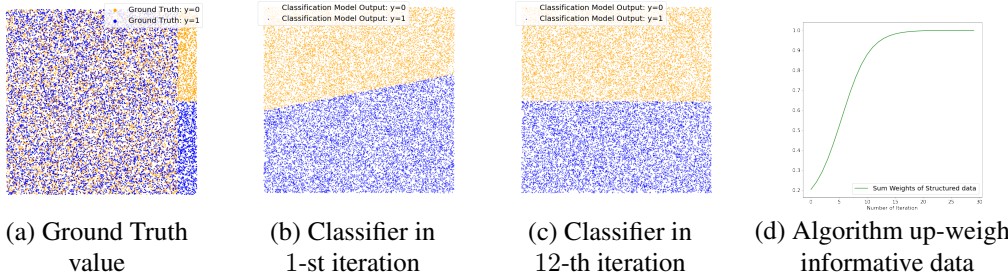

(a) Ground Truth value

(b) Classifier in 1-st iteration

(c) Classifier in 12-th iteration

(d) Algorithm up-weighs informative data

Figure 2: Illustration of Algorithm 1. By up-weighing the informative datapoints, the algorithm progressively improves the classifier. d) shows the sum weight of all informative data over weight sum of all data, i.e $\frac{\sum_{i:x_i \in \Omega_I} \gamma_i}{\sum_i^n \gamma_i}$ ( See $\gamma$ in Algorithm 1).

## B   THEORETICAL ANALYSIS

In this section, we show auxiliary results culminating with the proof of the main result in Theorem 4. In this section, we will use the following well-known result repeatedly.

**Lemma 1** (Hoeffding's Inequality). *Let $Z_1, ..., Z_n$ be independent bounded random variables with $Z_i \in [a, b]$ for all $i$, where $-\infty < a < b < \infty$. Then for all $t > 0$:*

$$\mathbb{P}(\frac{1}{n}|\sum_{i=1}^n Z_i - \mathbb{E}[Z_i]| \geq t) \leq 2e^{-\frac{2nt^2}{(b-a)^2}} \tag{9}$$

The next lemma provides an upper bound on the empirical risk $R_S(f^*)$, if enough samples are available.

**Lemma 2.** *Consider a set of samples $S = \{(x_1, y_1), ..., (x_n, y_n)\}$ drawn i.i.d. from the Noisy Generative Process and $f^*$ in the hypothesis class $\mathcal{F}$ satisfying $f(x) \in \{-1, +1\}$. If:*

$$n \geq \frac{3 \log(\frac{1}{\delta})}{\epsilon^2 \alpha^2}$$

*Then we have with probability at least $1 - \delta$ :*

$$R_S(f^*) \leq \frac{1}{2}(1 - \alpha) + \alpha \varepsilon \tag{10}$$

**Proof**: By definition:

$$R(f) = \mathbb{E}_{(x,y) \sim \mathcal{D}_\alpha}[\mathbb{1} f(x) \neq y] \tag{11}$$

and the empirical risk

$$R_S(f) = \frac{1}{n} \sum_{i=1}^n \mathbb{1}\{f(x_i) \neq y_i\} \tag{12}$$

Note that $\mathbb{1}\{f(x) \neq y\}$ is bounded in the interval $[0, 1]$ and given $f^* \in \mathcal{F}$, $\mathbb{1}\{f(x_i) \neq y_i\}, i \in [n]$ form a set of $n$ independent random variables. By setting $b - a = 1$, $t = \alpha \epsilon$ in Equation 9, the choice of $n$ ensures that $\frac{-2nt^2}{(b-a)^2} \leq 6 \log(\delta)$. Thus

$$\mathbb{P}_{S_n \sim \mathcal{D}_\alpha}[|R_S(f^*) - R(f^*)| \geq \epsilon \alpha] \leq \delta.$$

where we have

$$R(f^*) = \mathbb{E}_{(x,y)\sim\mathcal{D}_\alpha}[\mathbb{1}f(x) \neq y]$$

$$= \underbrace{\mathbb{E}_{(x,y)\sim\mathcal{D}_\alpha}[\mathbb{1}\{f^*(x) \neq y\}|x \in \Omega_U]\mathbb{P}[x \in \Omega_U]}_{\frac{1}{2}\mathbb{P}[x \in \Omega_U]: \text{ Since } y \text{ is labeled by coin flipping in } \Omega_U}$$

$$+ \underbrace{\mathbb{E}_{(x,y)\sim\mathcal{D}_\alpha}[\mathbb{1}\{f^*(x) \neq y\}|x \in \Omega_I]\mathbb{P}[x \in \Omega_I]}_{0: \text{ Since } y \text{ is labeled by } f^* \text{ with 0 Bayes Risk in } \Omega_I} \tag{13}$$

$$= \frac{1}{2}(1 - \alpha)$$

This way we have:

$$\mathbb{P}_{S_n\sim\mathcal{D}_\alpha}[|R_S(f^*) - \frac{1}{2}(1 - \alpha)| \geq \epsilon\alpha] \leq \delta.$$

which implies that Equation 10 holds with probability at least $1 - \delta$. $\qquad\square$

The proof of Lemma 2 relies on the independence between $\mathbb{1}f(x_i) \neq y_i$ across different pairs $(x_i, y_i) \in S_n$. However, there is no guarantee that such independence holds for the empirical minimizer $f^*_{S_n} = \arg\min_{f\in\mathcal{F}} R_S(f)$..

In order to prove Theorem 1, we use the growth function in Definition 7 for sets of size $n$ and the following symmetrization lemma. Here we show its proof for completeness; it can be found in different sources (Mohri et al., 2018; Blum et al., 2016).

**Lemma 3.** *Suppose $S_n = \{(x_1, y_1), ..., (x_n, y_n)\}$ are i.i.d sampled , $L(f, x, y) \in [0, b]$ and $L_{S_n} = \frac{1}{n}\sum_{i=1}^n L(f, x_i, y_i)$. If*

$$nt^2 \geq 2b^2$$

*then we have:*

$$\mathbb{P}_{S_n\sim\mathcal{D}_\alpha}[\sup_{f\in\mathcal{F}}|L_{S_n}(f) - L(f)| \geq t] \leq 2\mathbb{P}_{S_n,S'_n\sim\mathcal{D}_\alpha}[\sup_{f\in\mathcal{F}}|L_{S_n}(f) - L_{S'_n}(f)| \geq \frac{t}{2}].$$

**Proof**: Since $|L_{S_n}(f) - L(f)| \geq t$ and $|L_{S'_n}(f) - L(f)| \leq \frac{t}{2}$ implies $|L_{S_n} - L_{S'_n}| \geq \frac{t}{2}$ thus

$$\mathbb{1}_{\sup_{f\in\mathcal{F}}|L_{S_n}(f)-L(f)|\geq t}\mathbb{1}_{\sup_{f\in\mathcal{F}}|L_{S'_n}(f)-L(f)|\leq\frac{t}{2}}$$

$$\leq \mathbb{1}_{\sup_{f\in\mathcal{F}}|L_{S_n}(f)-L_{S'_n}(f)|\geq\frac{t}{2}} \tag{14}$$

Taking expectation w.r.t $S_n \sim \mathcal{D}_\alpha$ and $S'_n \sim \mathcal{D}_\alpha$ we have

$$\mathbb{P}_{S_n\sim\mathcal{D}_\alpha}[\sup_{f\in\mathcal{F}}|L_{S_n}(f) - L(f)| \geq t]\mathbb{P}_{S'_n\sim\mathcal{D}_\alpha}[\sup_{f\in\mathcal{F}}|L_{S'_n}(f) - L(f)| \leq \frac{t}{2}]$$

$$\leq \mathbb{P}_{S_n,S'_n\sim\mathcal{D}_\alpha}[\sup_{f\in\mathcal{F}}|L_{S_n}(f) - L_{S'_n}(f)| \geq \frac{t}{2}] \tag{15}$$

Next we lower bound $\mathbb{P}[\sup_{f\in\mathcal{F}}|L_{S'_n}(f) - L(f)| \geq \frac{t}{2}]$. Since $L(f, x, y) \in [0, b]$ and $Var(L(f, x, y)) \leq \frac{b^2}{4}$, we have:

$$\mathbb{P}_{S'_n\sim\mathcal{D}_\alpha}[\sup_{f\in\mathcal{F}}|L_{S'_n}(f) - L(f)| \leq \frac{t}{2}] \leq \frac{4Var(L_{S_n})}{nt^2} \leq \frac{1}{2}$$

Given the choice of $n \geq 2b^2$, we have $\mathbb{P}[\sup_{f\in\mathcal{F}}|L_{S'_n}(f) - L(f)| \geq \frac{t}{2}] \geq \frac{1}{2}$ which implies

$$\mathbb{P}_{S_n\sim\mathcal{D}_\alpha}[\sup_{f\in\mathcal{F}}|L_{S_n}(f) - L(f)| \geq t] \leq 2\mathbb{P}_{S_n,S'_n\sim\mathcal{D}_\alpha}[\sup_{f\in\mathcal{F}}|L_{S_n}(f) - L_{S'_n}(f)| \geq \frac{t}{2}].$$

**Theorem 1.** *Under Assumption 1, given a set of samples $S_n = \{(x_1, y_1), ..., (x_n, y_n)\}$ drawn i.i.d. from the Noisy Generative Process and*

$$f^*_{S_n} = \arg\min_{f\in\mathcal{F}} R_S(f),$$

*suppose*

$$n \geq \frac{32 \left[ \log(4|\mathcal{G}_{\mathcal{H}}(2n)|) + \log(\frac{1}{\delta}) \right]}{\epsilon^2 \alpha^2}$$

*Then with probability at least $1 - 2\delta$:*

$$R(f_{S_n}^*) \leq \frac{1}{2}(1 - \alpha) + 2\epsilon\alpha.$$

*Furthermore,*

$$\mathbb{P}_{x \sim \mathcal{D}_I}[f^*(x) \neq f_{S_n}^*(x)] \leq 2\epsilon$$

*Proof.* : We first bound the probability of the event that $R(f_{S_n}^*) \leq \frac{1}{2} - (1 - 2\epsilon)\alpha$.

We start with bounding $\mathbb{P}_{S_n, S_n' \sim \mathcal{D}_\alpha}[|R_{S_n}(f) - R(f) + R(f) - R_{S_n'}(f)| \geq \frac{t}{2}]$ for a fixed $f$. Since $|R_{S_n}(f) - R(f) + R(f) - R_{S_n'}(f)| \geq \frac{t}{2}$ implies $|R_{S_n}(f) - R(f)| \geq \frac{t}{4}$ or $|R(f) - R_{S_n'}(f)| \geq \frac{t}{4}$ we have

$$\mathbb{P}_{S_n, S_n' \sim \mathcal{D}_\alpha}[|R_{S_n}(f) - R(f) + R(f) - R_{S_n'}(f)| \geq \frac{t}{2}]$$

$$\leq \mathbb{P}_{S_n \sim \mathcal{D}_\alpha}[|R_{S_n}(f) - R(f)| \geq \frac{t}{4}] + \mathbb{P}_{S_n' \sim \mathcal{D}_\alpha}[|R_{S_n'}(f) - R(f)| \geq \frac{t}{4}] \tag{16}$$

$$\leq 2e^{-\frac{nt^2}{32}}$$

Let $\mathcal{F}_{S_n, S_n'} = \{f(x) : f \in \mathcal{F}, x \in S_n \cup S_n'\} \subseteq \{+1, -1\}^{2n}$ with Lemma 3 and Eq 16 we have:

$$\mathbb{P}_{S_n \sim \mathcal{D}_\alpha}[\sup_{f \in \mathcal{F}} |R_{S_n}(f) - R(f)| \geq t]$$

$$\leq 2\mathbb{P}_{S_n, S_n' \sim \mathcal{D}_\alpha}[\sup_{f \in \mathcal{F}} |R_{S_n}(f) - R_{S_n'}(f)| \geq \frac{t}{2}].$$

$$= 2\mathbb{P}_{S_n, S_n' \sim \mathcal{D}_\alpha}[\sup_{f \in \mathcal{F}_{S_n, S_n'}} |R_{S_n}(f) - R_{S_n'}(f)| \geq \frac{t}{2}]$$

$$\leq 2 \sum_{f \in \mathcal{F}_{S_n, S_n'}} \mathbb{P}[|R_{S_n}(f) - R_{S_n'}(f)| \geq \frac{t}{2}] \tag{17}$$

$$= 2 \sum_{f \in \mathcal{F}_{S_n, S_n'}} \mathbb{P}[|R_{S_n}(f) - R(f) + R(f) - R_{S_n'}(f)| \geq \frac{t}{2}]$$

$$\leq 4|\mathcal{F}_{S_n', S_n}|e^{-\frac{nt^2}{32}}$$

$$\leq 4\mathcal{G}_{\mathcal{F}}(2n)e^{-\frac{nt^2}{32}}$$

By setting $t = \alpha\epsilon$ and $n \geq \frac{32(4\log(\mathcal{G}_{\mathcal{F}}(2n)) + \log(\frac{1}{\delta}))}{\alpha^2 \epsilon^2}$ we have with probability of at least $1 - \delta$:

$$R(f_{S_n}^*) - R_{S_n}(f_{S_n}^*) \leq \frac{\alpha\epsilon}{2}$$

Next we apply the fact that $f_{S_n}^* = \arg\min_{f \in \mathcal{F}} R_S(f)$. Since $R_{S_n}(f^*) \geq R_{S_n}(f_{S_n}^*)$, we have:

$$R(f_{S_n}^*) \leq \frac{\alpha\epsilon}{2} + R_{S_n}(f_{S_n}^*) \leq \frac{\alpha\epsilon}{2} + R_{S_n}(f^*)$$

Since $R_{S_n}(f^*) \leq \frac{1}{2}(1 - \alpha) + \epsilon\alpha$ with failure probability at most $\delta$ (Lemma 2), we have with probability at least $1 - 2\delta$:

$$R(f_{S_n}^*) \leq \frac{1}{2}(1 - \alpha) + 2\epsilon\alpha.$$

Next we prove the claim that:

$$\mathbb{P}_{x \sim \mathcal{D}_I}[f^*(x) \neq f_{S_n}^*(x)] \leq 2\epsilon.$$

Since $R(f_{S_n}^*) \leq \frac{1}{2}(1-\alpha) + 2\epsilon\alpha$:

$$
\begin{aligned}
R(f_{S_n}^*) =& \mathbb{E}_{(x,y)\sim\mathcal{D}_\alpha}[\mathbb{1}\{f_{S_n}^*(x) \neq y\}] \\
=& \mathbb{P}_{(x,y)\sim\mathcal{D}_\alpha}[\mathbb{1}\{f_{S_n}^*(x) \neq y\}] \\
=& \underbrace{\mathbb{P}_{(x,y)\sim\mathcal{D}_\alpha}[\mathbb{1}\{f_{S_n}^*(x) \neq y\}|x \in \Omega_U]}_{\frac{1}{2}} \underbrace{\mathbb{P}_{(x,y)\sim\mathcal{D}_\alpha}[x \in \Omega_U]}_{1-\alpha} \\
& + \underbrace{\mathbb{P}_{(x,y)\sim\mathcal{D}_\alpha}[\mathbb{1}\{f_{S_n}^*(x) \neq y\}|x \in \Omega_I]}_{\mathbb{P}_{(x,y)\sim\mathcal{D}_\alpha}[\mathbb{1}\{f_{S_n}^*(x)\neq f^*(x)\}|x\in\Omega_I]} \underbrace{\mathbb{P}_{(x,y)\sim\mathcal{D}_\alpha}[x \in \Omega_I]}_{\alpha} \\
=& \frac{1}{2}(1-\alpha) + \alpha\mathbb{P}_{x\sim\mathcal{D}_\alpha}[\mathbb{1}\{f_{S_n}^*(x) \neq f^*(x)\}|x \in \Omega_I] \\
\leq& \frac{1}{2}(1-\alpha) + 2\epsilon\alpha \\
\Longrightarrow& \mathbb{P}_{x\sim\mathcal{D}_\alpha}[\mathbb{1}\{f_{S_n}^*(x) \neq f^*(x)\}|x \in \Omega_I] \leq 2\epsilon
\end{aligned}
\tag{18}
$$

$\square$

Next we analyze the weighted risk in Definition 6 and its empirical version. In particular, we are interested in the case where $\theta > \alpha$, which implies $\frac{\theta}{\alpha} \geq \frac{1-\theta}{1-\alpha}$. For a given pair $(x_i, y_i)$, we have:

$$
\left\{ \frac{\theta}{\alpha}\mathbb{1}\{f(x_i) \neq y_i\}\mathbb{1}\{g(x_i) > 0\} + \frac{1-\theta}{1-\alpha}\mathbb{1}\{f(x_i) = y_i\}\mathbb{1}\{g(x_i) \leq 0\} \right\} \leq \frac{\theta}{\alpha}
$$

.

It is useful to write down explicitly the following decomposition of $W(g, f, \theta)$:

$$
\begin{aligned}
W(g, f, \theta) =& \mathbb{E}_{(x,y)\sim\mathcal{D}_\alpha}\left[\frac{\theta}{\alpha}\mathbb{1}\{f(x) \neq y\}\mathbb{1}\{g(x) > 0\} + \frac{1-\theta}{1-\alpha}\mathbb{1}\{f(x) = y\}\mathbb{1}\{g(x) \leq 0\}\right] \\
=& \mathbb{E}_{(x,y)\sim\mathcal{D}_\alpha}\left[\frac{\theta}{\alpha}\mathbb{1}\{f(x) \neq y\}\mathbb{1}\{g(x) > 0\}\mathbb{1}\{g^*(x) > 0\}\right] \\
& + \mathbb{E}_{(x,y)\sim\mathcal{D}_\alpha}\left[\frac{\theta}{\alpha}\mathbb{1}\{f(x) \neq y\}\mathbb{1}\{g(x) > 0\}\mathbb{1}\{g^*(x) \leq 0\}\right] \\
& + \mathbb{E}_{(x,y)\sim\mathcal{D}_\alpha}\left[\frac{1-\theta}{1-\alpha}\mathbb{1}\{f(x) = y\}\mathbb{1}\{g(x) \leq 0\}\mathbb{1}\{g^*(x) \leq 0\}\right] \\
& + \mathbb{E}_{(x,y)\sim\mathcal{D}_\alpha}\left[\frac{1-\theta}{1-\alpha}\mathbb{1}\{f(x) = y\}\mathbb{1}\{g(x) \leq 0\}\mathbb{1}\{g^*(x) > 0\}\right] \\
=& \frac{\theta}{\alpha}\mathbb{P}_{(x,y)\sim\mathcal{D}_\alpha}[f(x) \neq y|g(x) > 0, g^*(x) > 0]\mathbb{P}_{x\sim\mathcal{D}_\alpha}[g(x) > 0, g^*(x) > 0] \\
& + \frac{\theta}{\alpha}\mathbb{P}_{(x,y)\sim\mathcal{D}_\alpha}[f(x) \neq y|g(x) > 0, g^*(x) \leq 0]\mathbb{P}_{x\sim\mathcal{D}_\alpha}[g(x) > 0, g^*(x) \leq 0] \\
& + \frac{1-\theta}{1-\alpha}\mathbb{P}_{(x,y)\sim\mathcal{D}_\alpha}[f(x) = y|g(x) \leq 0, g^*(x) \leq 0] \\
& + \frac{1-\theta}{1-\alpha}\mathbb{P}_{(x,y)\sim\mathcal{D}_\alpha}[f(x) = y|g(x) \leq 0, g^*(x) > 0]\mathbb{P}_{x\sim\mathcal{D}_\alpha}[g(x) \leq 0, g^*(x) > 0] \\
=& \frac{\theta}{\alpha}\mathbb{P}_{(x,y)\sim\mathcal{D}_\alpha}[f(x) \neq y|g(x) > 0, g^*(x) > 0]\mathbb{P}_{x\sim\mathcal{D}_\alpha}[g(x) > 0, g^*(x) > 0] \\
& + \frac{\theta}{2\alpha}\mathbb{P}_{x\sim\mathcal{D}_\alpha}[g(x) > 0, g^*(x) \leq 0] \\
& + \frac{1-\theta}{2(1-\alpha)}\mathbb{P}_{x\sim\mathcal{D}_\alpha}[g(x) \leq 0, g^*(x) \leq 0] \\
& + \frac{1-\theta}{1-\alpha}\mathbb{P}_{(x,y)\sim\mathcal{D}_\alpha}[f(x) = y|g(x) \leq 0, g^*(x) > 0]\mathbb{P}_{x\sim\mathcal{D}_\alpha}[g(x) \leq 0, g^*(x) > 0]
\end{aligned}
\tag{19}
$$

One can see that if $\theta > \alpha$, then $g(x)$ should reject all $x \in \Omega_U$ since $\frac{\theta}{\alpha} > \frac{1-\theta}{1-\alpha}$. For the informative partition $x \in \Omega_I$, the decomposition suggests acceptance of all $x$ such that $f(x) = y$ and rejection of $f(x) \neq y$. This gives the optimal value $\frac{1-\theta}{2(1-\alpha)}\mathbb{P}_{x\sim\mathcal{D}_\alpha}[g^*(x) \leq 0]$. Note that the optimal value of $W(g, f, \theta)$ is not necessarily achieved at $(f^*, g^*)$. For any fixed $f$, the optimal value of $W(g, f, \theta)$ is achived for $g$ as long as $g$ accepts all datapoints in $\Omega_I$ that are correctly classified by $f$, and rejects all mistakes made by $f$ in $\Omega_I$ (as well as all datapoints in $\Omega_U$).

**Lemma 4.** *Given a set of samples* $S_n = \{(x_1, y_1), ..., (x_n, y_n)\}$ *drawn i.i.d. from the Noisy Generative Process. Suppose* $\theta \in (\alpha, 1), \alpha \in (0, 1), \epsilon \leq 0.1$, *and:*

$$n \geq \frac{2\theta^2 \log(\frac{1}{\delta})}{\epsilon^2 \alpha^4}$$

*Then with probability at least* $1 - \delta$ *we have:*

$$|W_{S_n}(g^*, f^*, \theta) - W(g^*, f^*, \theta)| \leq \alpha\epsilon$$

*Proof.* The following holds for a given $(x_i, y_i)$:

$$\left\{\frac{\theta}{\alpha}\mathbb{1}\{f(x_i) \neq y_i\}\mathbb{1}\{g(x_i) > 0\} + \frac{1-\theta}{1-\alpha}\mathbb{1}\{f(x_i) = y_i\}\mathbb{1}\{g(x_i) \leq 0\}\right\} \leq \frac{\theta}{\alpha}.$$

Since $\theta < \frac{1}{2}$, we can apply Hoeffding's inequality in Lemma 1 with $a = 0, b = \frac{\theta}{\alpha}$. For fixed $g, f \in \mathcal{F}, \mathbb{1}\{f(x_i) \neq y_i\}, i \in [n]$ are set of $n$ independent random variables. By setting $b - a = \frac{\theta}{\alpha}$, $t = \alpha\epsilon$ in Equation 9, the choice of $n$ ensures that $\frac{-2nt^2}{(b-a)^2} \leq 6\log(\delta)$. Therefore:

$$\mathbb{P}_{S_n\sim\mathcal{D}_\alpha}[|W_{S_n}(g^*, f^*, \theta) - W(g^*, f^*, \theta)| \geq \epsilon\alpha] \leq \delta.$$

$\square$

Recall the empirical minimizer of $W_{S_n}(g, f^*_{S_n}, \theta)$:

$$g_{S_n}^* = \arg\min_{g\in\mathcal{H}} W_{S_n}(g, f^*_{S_n}, \theta).$$

Since $f^*_{S_n}$ and $g^*_{S_n}$ are data-dependent, it follows that $W_i(g^*_{S_n}, f^*_{S_n}, \theta), i \in [n]$ are not independent draws. Our strategy is to use similar logic as in the proof of the Theorem 1. We begin by finding an upper bound on the probability of $W_{S_n}(g, f, \theta)$ deviating from its expectation. Then we take a union bound of all the possible pair of $(g, f) \in \mathcal{H} \times \mathcal{F}$ restricted to $S_n$.

**Theorem 2.** *Consider a set of samples* $S_n = \{(x_1, y_1), ..., (x_n, y_n)\}$ *drawn i.i.d. from the Noisy Generative Process and*

$$f^*_{S_n} = \arg\min_{f\in\mathcal{F}} R_S(f), g_{S_n}^* = \arg\min_{g\in\mathcal{H}} W_{S_n}(g, f^*_{S_n}, \theta),$$

*If* $\theta \in (\alpha, 1), \alpha \in (0, 1)$ *and*

$$n \geq \frac{32\theta^2 \left(log(4|\mathcal{G}_\mathcal{H}(2n)||\mathcal{G}_\mathcal{F}(2n)|) + \log(\frac{1}{\delta})\right)}{\epsilon^2 \alpha^4}$$

*then with probability at least* $1 - \delta$:

$$|W(g^*_{S_n}, f^*_{S_n}, \theta) - W_{S_n}(g^*_{S_n}, f^*_{S_n}, \theta)| \leq \epsilon\alpha.$$

*Proof.* First we apply Lemma 3 with $L_{S_n}(f,g) = W_{S_n}(g,f,\theta)$, $b = \frac{\theta}{\alpha}$ and $t = \alpha\epsilon$, and also using the condition $n \geq \frac{32\theta^2 log(2|\mathcal{G}_\mathcal{H}(2n)||\mathcal{G}_\mathcal{F}(2n)|) + \log(\frac{1}{\delta})}{\epsilon^2\alpha^4} \geq \frac{2\theta^2}{\epsilon^2\alpha^4}$ in the lemma. Then it follows:

$$\mathbb{P}_{S_n \sim \mathcal{D}_\alpha}[\sup_{g \in \mathcal{H}} |W_{S_n}(g, f^*_{S_n}, \theta) - W(g, f^*_{S_n}, \theta)| \geq \alpha\epsilon]$$

$$\leq 2\mathbb{P}_{S_n, S'_n \sim \mathcal{D}_\alpha}[\sup_{g \in \mathcal{H}} |W_{S_n}(g, f^*_{S_n}, \theta) - W_{S'_n}(g, f^*_{S_n}, \theta)| \geq \frac{\alpha\epsilon}{2}]$$

$$\leq 2\mathbb{P}_{S_n, S'_n \sim \mathcal{D}_\alpha}[\sup_{g \in \mathcal{H}, f \in \mathcal{F}} |W_{S_n}(g, f, \theta) - W_{S'_n}(g, f, \theta)| \geq \frac{\alpha\epsilon}{2}]$$

$$\leq 2\mathbb{P}_{S_n, S'_n \sim \mathcal{D}_\alpha}[\sup_{g \in \mathcal{H}_{S_n, S'_n}, f \in \mathcal{F}_{S_n, S'_n}} |W_{S_n}(g, f, \theta) - W_{S'_n}(g, f, \theta)| \geq \frac{\alpha\epsilon}{2}]$$

$$\leq 2 \sum_{g \in \mathcal{H}_{S_n, S'_n}, f \in \mathcal{F}_{S_n, S'_n}} \mathbb{P}_{S_n, S'_n \sim \mathcal{D}_\alpha}[|W_{S_n}(g, f, \theta) - W_{S'_n}(g, f, \theta)| \geq \frac{\alpha\epsilon}{2}]$$

$$\leq 2 \sum_{g \in \mathcal{H}_{S_n, S'_n}, f \in \mathcal{F}_{S_n, S'_n}} \mathbb{P}_{S_n, S'_n \sim \mathcal{D}_\alpha}[|W_{S_n}(g, f, \theta) - W(g, f, \theta) + W(g, f, \theta) - W_{S'_n}(g, f, \theta)| \geq \frac{\alpha\epsilon}{2}]$$

$$\leq 4|\mathcal{H}_{S_n, S'_n}||\mathcal{F}_{S_n, S'_n}|\mathbb{P}_{S_n \sim \mathcal{D}_\alpha}[|W_{S_n}(g, f, \theta) - W(g, f, \theta)| \geq \frac{\alpha\epsilon}{4}]$$

$$\leq 4\mathcal{G}_\mathcal{H}(2n)\mathcal{G}_\mathcal{F}(2n)e^{-\frac{n\alpha^2\epsilon^2}{32\frac{\theta^2}{\alpha^2}}}$$

$$= 4\mathcal{G}_\mathcal{H}(2n)\mathcal{G}_\mathcal{F}(2n)e^{-\frac{n\alpha^4\epsilon^2}{32\theta^2}}$$

$$\tag{20}$$

□

Next we bound the FPR and FNR of $g^*_{S_n}$ leveraging on the fact that $W_{S_n}(g, f^*_{S_n})$ is sufficiently small.

**Theorem 3.** *Under Assumption 1, given a set of samples $S_n = \{(x_1, y_1), ..., (x_n, y_n)\}$ drawn i.i.d. from Noisy Generative Process and*

$$f^*_{S_n} = \arg\min_{f \in \mathcal{F}} R_S(f), g_{S_n}^* = \arg\min_{g \in \mathcal{H}} W_{S_n}(g, f^*_{S_n}, \theta),$$

*if $\theta \in (\alpha, 1), \alpha \in (0, 1)$ and*

$$n \geq \frac{32\theta^2 \left[ log(4|\mathcal{G}_\mathcal{H}(2n)||\mathcal{G}_\mathcal{F}(2n)|) + \log(\frac{5}{\delta}) \right]}{\epsilon^2\alpha^4}$$

*We have*

    *I) with probability at least $1 - \delta$*

$$W(g^*_{S_n}, f^*_{S_n}, \theta) \leq 2\epsilon\theta + 2\alpha\epsilon + \frac{(1-\theta)}{2}$$

    *II) if $\theta = 2c\alpha$ with $\frac{1}{2} < c < \frac{1+\alpha}{4\alpha}$, we have with probability at least $1 - \delta$*

$$\mathbb{P}[g^*_{S_n}(x) \geq 0 | g^*(x) \leq 0] \leq \frac{12c\alpha\epsilon}{2c-1} \textit{[False Positive]} \tag{21}$$

$$\mathbb{P}[g^*_{S_n}(x) < 0 | g^*(x) \geq 0] \leq 20c\epsilon \textit{[False Negative]} \tag{22}$$

**Proof of I)**

*Proof.*

$$W(g^*_{S_n}, f^*_{S_n}, \theta)$$

$$\leq \underbrace{W_{S_n}(g^*_{S_n}, f^*_{S_n}, \theta) + \alpha\epsilon}_{\text{with failure probability at most } \delta \text{ Theorem 2)}}$$

$$\leq \underbrace{W_{S_n}(g^*, f^*_{S_n}, \theta) + \alpha\epsilon}_{\text{Optimality of } g^*_{S_n}}$$

$$\leq \underbrace{W(g^*, f^*_{S_n}, \theta) + 2\alpha\epsilon}_{\text{With failure probability at most } \delta \text{ (Lemma 4)}}$$

$$= \mathbb{E}_{(x,y)\sim\mathcal{D}_\alpha} \left[ \frac{\theta}{\alpha} \mathbb{1}\{f^*_{S_n}(x) \neq y\} \mathbb{1}\{g^*(x) > 0\} \right]$$

$$+ \mathbb{E}_{(x,y)\sim\mathcal{D}_\alpha} \left[ \frac{1-\theta}{1-\alpha} \mathbb{1}\{f^*_{S_n}(x) = y\} \mathbb{1}\{g^*(x) \leq 0\} \right] + 2\alpha\epsilon$$

$$= \mathbb{E}_{(x,y)\sim\mathcal{D}_\alpha} \left[ \frac{\theta}{\alpha} \mathbb{1}\{f^*_{S_n}(x) \neq y\} | g^*(x) > 0 \right] \mathbb{P}_{x\sim\mathcal{D}_\alpha} [g^*(x) \geq 0]$$

$$+ \mathbb{E}_{(x,y)\sim\mathcal{D}_\alpha} \left[ \frac{1-\theta}{1-\alpha} \mathbb{1}\{f^*_{S_n}(x) = y\} | g^*(x) \leq 0 \right] \mathbb{P}_{x\sim\mathcal{D}_\alpha} [g^*(x) \leq 0] + 2\alpha\epsilon$$

$$= \mathbb{E}_{(x,y)\sim\mathcal{D}_\alpha} \left[ \theta \mathbb{1}\{f^*_{S_n}(x) \neq y\} | g^*(x) > 0 \right] + \mathbb{E}_{(x,y)\sim\mathcal{D}_\alpha} \left[ (1-\theta) \mathbb{1}\{f^*_{S_n}(x) = y\} | g^*(x) \leq 0 \right]$$

$$= \theta \underbrace{\mathbb{P}_{x\sim\mathcal{D}_I}[f^*_{S_n} \neq f^*(x)]}_{\leq 2\epsilon \text{ with failure prob at most } \delta \text{ (Thm 1)}}$$

$$+ \underbrace{(1-\theta)\mathbb{P}_{x\sim\mathcal{D}_U}[f^*_{S_n} = y]}_{=0.5 \text{ since all y's in } \Omega_U \text{ are coin flipping}} + 2\alpha\epsilon$$

$$\leq 2\epsilon\theta + 2\alpha\epsilon + \frac{(1-\theta)}{2}$$

$$\tag{23}$$

$$\square$$

**Proof of II)**:

*Proof.*

$$W(g^*_{S_n}, f^*_{S_n}, \theta)$$

$$= \mathbb{E}_{(x,y)\sim\mathcal{D}_\alpha} \left[ \frac{\theta}{\alpha} \mathbb{1}\{f^*_{S_n}(x) \neq y\} \mathbb{1}\{g^*_{S_n}(x) > 0\} \right]$$

$$+ \mathbb{E}_{(x,y)\sim\mathcal{D}_\alpha} \left[ \frac{1-\theta}{1-\alpha} \mathbb{1}\{f^*_{S_n}(x) = y\} \mathbb{1}\{g^*_{S_n}(x) \leq 0\} \right] \tag{24}$$

$$= \underbrace{\mathbb{E}_{(x,y)\sim\mathcal{D}_\alpha} \left[ \frac{\theta}{\alpha} \mathbb{1}\{f^*_{S_n}(x) \neq y\} \mathbb{1}\{g^*_{S_n}(x) > 0\} | g^*(x) \geq 0 \right] \mathbb{P}_{x\sim\mathcal{D}_\alpha}[g^*(x) \geq 0]}_{\text{Term 1}}$$

$$+ \underbrace{\mathbb{E}_{(x,y)\sim\mathcal{D}_\alpha} \left[ \frac{\theta}{\alpha} \mathbb{1}\{f^*_{S_n}(x) \neq y\} \mathbb{1}\{g^*_{S_n}(x) > 0\} | g^*(x) \leq 0 \right] \mathbb{P}_{x\sim\mathcal{D}_\alpha}[g^*(x) \leq 0]}_{\text{Term 2}}$$

$$+ \underbrace{\mathbb{E}_{(x,y)\sim\mathcal{D}_\alpha} \left[ \frac{1-\theta}{1-\alpha} \mathbb{1}\{f^*_{S_n}(x) = y\} \mathbb{1}\{g^*_{S_n}(x) \leq 0\} | g^*(x) > 0 \right] \mathbb{P}_{x\sim\mathcal{D}_\alpha}[g^*(x) \geq 0]}_{\text{Term 3}}$$

$$+ \underbrace{\mathbb{E}_{(x,y)\sim\mathcal{D}_\alpha} \left[ \frac{1-\theta}{1-\alpha} \mathbb{1}\{f^*_{S_n}(x) = y\} \mathbb{1}\{g^*_{S_n}(x) \leq 0\} | g^*(x) \leq 0 \right] \mathbb{P}_{x\sim\mathcal{D}_\alpha}[g^*(x) \leq 0]}_{\text{Term 4}}$$

Next we lower bound Term $1, 2, 3, 4$ individually and apply upper bound on $W(g^*_{S_n}, f^*_{S_n}, \theta)$ in Eq 23 for the proof of the statement.

**Term 1**:

$$
\begin{aligned}
&\mathbb{E}_{(x,y)\sim\mathcal{D}_\alpha}\left[\frac{\theta}{\alpha}\mathbb{1}\{f^*_{S_n}(x)\neq y\}\mathbb{1}\{g^*_{S_n}(x)>0\}|g^*(x)\geq 0\right]\mathbb{P}_{x\sim\mathcal{D}_\alpha}[g^*(x)\geq 0] \\
=&\mathbb{E}_{(x,y)\sim\mathcal{D}_\alpha}\left[\mathbb{1}\{f^*_{S_n}(x)\neq y\}\mathbb{1}\{g^*_{S_n}(x)>0\}|g^*(x)\geq 0\right]\theta \\
=&\theta\mathbb{E}_{(x,y)\sim\mathcal{D}_\alpha}\left[\mathbb{1}\{f^*_{S_n}(x)\neq y\}|g^*_{S_n}(x)>0,g^*(x)\geq 0\right]\mathbb{P}_{x\sim\mathcal{D}_\alpha}[g^*_{S_n}(x)>0|g^*(x)\geq 0] \\
\geq&0
\end{aligned}
\tag{25}
$$

**Term 2**:

$$
\begin{aligned}
&\mathbb{E}_{(x,y)\sim\mathcal{D}_\alpha}\left[\frac{\theta}{\alpha}\mathbb{1}\{f^*_{S_n}(x)\neq y\}\mathbb{1}\{g^*_{S_n}(x)>0\}|g^*(x)\leq 0\right]\mathbb{P}_{x\sim\mathcal{D}_\alpha}[g^*(x)\leq 0] \\
=&\frac{\theta(1-\alpha)}{\alpha}\mathbb{E}_{(x,y)\sim\mathcal{D}_\alpha}\left[\mathbb{1}\{f^*_{S_n}(x)\neq y\}\mathbb{1}\{g^*_{S_n}(x)>0\}|g^*(x)\leq 0\right] \\
=&\frac{\theta(1-\alpha)}{\alpha}\underset{\mathbb{P}_{x\sim\mathcal{D}_U}[f^*_{S_n}\neq y]=0.5 \text{ Since all } y \text{ are coin flipping if } g^*(x)\leq 0}{\mathbb{E}_{(x,y)\sim\mathcal{D}_\alpha}\left[\mathbb{1}\{f^*_{S_n}(x)\neq y\}|g^*_{S_n}(x)>0,g^*(x)\leq 0\right]}\mathbb{P}_{x\sim\mathcal{D}_\alpha}[g^*_{S_n}(x)>0|g^*(x)\leq 0] \\
=&\frac{\theta(1-\alpha)}{2\alpha}\mathbb{P}_{x\sim\mathcal{D}_\alpha}[g^*_{S_n}(x)>0|g^*(x)\leq 0]
\end{aligned}
\tag{26}
$$

**Term 3**:

$$
\begin{aligned}
&\mathbb{E}_{(x,y)\sim\mathcal{D}_\alpha}\left[\frac{1-\theta}{1-\alpha}\mathbb{1}\{f^*_{S_n}(x)=y\}\mathbb{1}\{g^*_{S_n}(x)\leq 0\}|g^*(x)>0\right]\mathbb{P}_{\mathcal{D}_\alpha}[g^*(x)\geq 0] \\
=&\frac{(1-\theta)\alpha}{1-\alpha}\mathbb{E}_{(x,y)\sim\mathcal{D}_\alpha}\left[\mathbb{1}\{f^*_{S_n}(x)=y\}\mathbb{1}\{g^*_{S_n}(x)\leq 0\}|g^*(x)>0\right] \\
=&\frac{(1-\theta)\alpha}{1-\alpha}\mathbb{P}_{(x,y)\sim\mathcal{D}_\alpha}\left[f^*_{S_n}(x)=y|g^*_{S_n}(x)\leq 0,g^*(x)>0\right]\mathbb{P}_{x\sim\mathcal{D}_\alpha}[g^*_{S_n}(x)\leq 0|g^*(x)>0]
\end{aligned}
\tag{27}
$$

By Theorem 1 we have with failure probability at most $2\delta$:

$$
\begin{aligned}
&\mathbb{P}_{x\sim\mathcal{D}_\alpha}[f^*_{S_n}\neq f^*(x)|g^*(x)\geq 0]\leq 2\epsilon \\
\Longrightarrow&\mathbb{P}_{x\sim\mathcal{D}_\alpha}[f^*_{S_n}\neq f^*(x),g^*_{S_n}(x)>0|g^*(x)>0]+\mathbb{P}_{x\sim\mathcal{D}_\alpha}[f^*_{S_n}\neq f^*(x),g^*_{S_n}(x)\leq 0|g^*(x)>0]\leq 2\epsilon \\
\Longrightarrow&\mathbb{P}_{x\sim\mathcal{D}_\alpha}[f^*_{S_n}\neq f^*(x)|g^*_{S_n}(x)\leq 0,g^*(x)>0]\mathbb{P}_{x\sim\mathcal{D}_\alpha}[g^*_{S_n}(x)\leq 0|g^*(x)>0]\leq 2\epsilon \\
\Longrightarrow&\left\{1-\mathbb{P}_{x\sim\mathcal{D}_\alpha}[f^*_{S_n}=f^*(x)|g^*_{S_n}(x)\leq 0,g^*(x)>0]\right\}\mathbb{P}_{x\sim\mathcal{D}_\alpha}[g^*_{S_n}(x)\leq 0|g^*(x)>0]\leq 2\epsilon \\
\Longrightarrow&\qquad\mathbb{P}_{x\sim\mathcal{D}_\alpha}[f^*_{S_n}=f^*(x)|g^*_{S_n}(x)\leq 0,g^*(x)>0]\mathbb{P}_{x\sim\mathcal{D}_\alpha}[g^*_{S_n}(x)\leq 0|g^*(x)>0] \\
&\qquad\geq\mathbb{P}_{x\sim\mathcal{D}_\alpha}[g^*_{S_n}(x)\leq 0|g^*(x)>0]-2\epsilon \\
\Longrightarrow&\qquad\frac{(1-\theta)\alpha}{1-\alpha}\mathbb{E}_{x\sim\mathcal{D}_\alpha}[\mathbb{1}\{f^*_{S_n}=f^*(x)\}|g^*_{S_n}(x)\leq 0,g^*(x)>0]\mathbb{P}_{x\sim\mathcal{D}_\alpha}[g^*_{S_n}(x)\leq 0|g^*(x)>0] \\
&\qquad\geq\frac{(1-\theta)\alpha}{1-\alpha}\left\{\mathbb{P}_{x\sim\mathcal{D}_\alpha}[g^*_{S_n}(x)\leq 0|g^*(x)>0]-2\epsilon\right\}
\end{aligned}
\tag{28}
$$

**Term 4**:

$$
\begin{aligned}
&\mathbb{E}_{(x,y)\sim\mathcal{D}_\alpha}\left[\frac{1-\theta}{1-\alpha}\mathbb{1}\{f^*_{S_n}(x)=y\}\mathbb{1}\{g^*_{S_n}(x)\leq 0\}|g^*(x)\leq 0\right]\mathbb{P}_{x\sim\mathcal{D}_\alpha}[g^*(x)\leq 0] \\
=&(1-\theta)\mathbb{E}_{(x,y)\sim\mathcal{D}_\alpha}\left[\mathbb{1}\{f^*_{S_n}(x)=y\}\mathbb{1}\{g^*_{S_n}(x)\leq 0\}|g^*(x)\leq 0\right] \\
=&(1-\theta)\underset{\forall x \quad s.t.g^*(x)\leq 0,y \text{ is coin flip}}{\mathbb{E}_{(x,y)\sim\mathcal{D}_\alpha}\left[\mathbb{1}\{f^*_{S_n}(x)=y\}|g^*_{S_n}(x)\leq 0,g^*(x)\leq 0\right]}\mathbb{P}_{x\sim\mathcal{D}_\alpha}[g^*_{S_n}(x)\leq 0|g^*(x)\leq 0] \\
=&\frac{1-\theta}{2}\left\{1-\mathbb{P}_{x\sim\mathcal{D}_\alpha}[g^*_{S_n}(x)>0|g^*(x)\leq 0]\right\}
\end{aligned}
\tag{29}
$$

Combining Equations 25, 26, 28, 29 we have:

$$W(g^*_{S_n}, f^*_{S_n}, \theta) \geq \left\{ \frac{\theta(1-\alpha)}{2\alpha} - \frac{1-\theta}{2} \right\} \mathbb{P}_{x \sim \mathcal{D}_\alpha}[g^*_{S_n}(x) > 0 | g^*(x) \leq 0] + \frac{1-\theta}{2}$$
$$+ \frac{\alpha(1-\theta)}{1-\alpha} \mathbb{P}_{x \sim \mathcal{D}_\alpha}[g^*_{S_n}(x) \leq 0 | g^*(x) > 0] - \frac{2(1-\theta)\alpha\epsilon}{1-\alpha} \tag{30}$$

and by part $I)$ in the theorem, we have with probability at least $1 - 3\delta$:

$$W(g^*_{S_n}, f^*_{S_n}, \theta) \leq 2\epsilon\theta + 2\alpha\epsilon + \frac{(1-\theta)}{2}$$

Combined with the failure probability $2\delta$ in the bound of **Term 3** we have the failure probability at most $5\delta$:

$$\left\{ \frac{\theta(1-\alpha)}{2\alpha} - \frac{1-\theta}{2} \right\} \mathbb{P}_{x \sim \mathcal{D}_\alpha}[g^*_{S_n}(x) > 0 | g^*(x) \leq 0] + \frac{1-\theta}{2}$$
$$+ \frac{\alpha(1-\theta)}{1-\alpha} \mathbb{P}_{x \sim \mathcal{D}_\alpha}[g^*_{S_n}(x) \leq 0 | g^*(x) > 0] - \frac{2(1-\theta)\alpha\epsilon}{1-\alpha}$$
$$\leq 2\epsilon\theta + 2\alpha\epsilon + \frac{(1-\theta)(1-\alpha)}{2} \tag{31}$$
$$\implies \left\{ \frac{\theta}{2\alpha} - \frac{1}{2} \right\} \mathbb{P}_{x \sim \mathcal{D}_\alpha}[g^*_{S_n}(x) > 0 | g^*(x) \leq 0]$$
$$+ \frac{\alpha(1-\theta)}{1-\alpha} \mathbb{P}_{x \sim \mathcal{D}_\alpha}[g^*_{S_n}(x) \leq 0 | g^*(x) > 0]$$
$$\leq 2\epsilon\theta + 2\alpha\epsilon + \frac{2(1-\theta)\alpha\epsilon}{1-\alpha}$$

By picking $\theta = 2c\alpha$ with $\frac{1}{2} < c < \frac{1+\alpha}{4\alpha}$ we can ensure that 1) $\frac{\theta}{2\alpha} - \frac{1}{2} > 0$; 2) $\frac{\alpha(1-\theta)}{1-\alpha} \geq \frac{\alpha}{2}$. Finally we have:

$$\left\{ c - \frac{1}{2} \right\} \mathbb{P}_{x \sim \mathcal{D}_\alpha}[g^*_{S_n}(x) > 0 | g^*(x) \leq 0] \leq 4c\alpha\epsilon + 4\alpha\epsilon$$
$$\implies \mathbb{P}_{x \sim \mathcal{D}_\alpha}[g^*_{S_n}(x) > 0 | g^*(x) \leq 0] \leq \frac{12c\alpha\epsilon}{2c-1} \tag{32}$$

and

$$\frac{\alpha(1-\theta)}{1-\alpha} \mathbb{P}_{x \sim \mathcal{D}_\alpha}[g^*_{S_n}(x) \leq 0 | g^*(x) > 0] \leq 4c\alpha\epsilon + \frac{2(1-\theta)\alpha\epsilon}{1-\alpha} + 2\alpha\epsilon$$
$$\implies \mathbb{P}_{x \sim \mathcal{D}_\alpha}[g^*_{S_n}(x) \leq 0 | g^*(x) > 0] \leq 20c\epsilon \tag{33}$$

$\square$

Given hypothesis class $\mathcal{H}$, one can bound the growth function $\mathcal{G}_\mathcal{H}(n)$ through its VC-dimension. Next we present the well known Sauer–Shelah Lemma below.

**Lemma 5** (Sauer–Shelah Lemma(See (Blum et al., 2016; Mohri et al., 2018; Sauer, 1972))). *Let $d_{vc}(\mathcal{H})$ be the VC-dimension of hypothesis class $\mathcal{H}$, for all $n \in \mathbb{N}$,*

$$\mathcal{G}_\mathcal{H}(n) \leq \sum_{i=0}^{d_{vc}} \binom{n}{i} \leq \left( \frac{en}{d_{vc}(\mathcal{H})} \right)^{d_{vc}(\mathcal{H})}$$

Now we are ready to prove the main result.

**Theorem 4.** *Under Assumption 1, given a set of samples $S_n = \{(x_1, y_1), ..., (x_n, y_n)\}$ drawn i.i.d. from the Noisy Generative Process, hypothesis class $\mathcal{H}$ with VC-dimension $d_{vc}(\mathcal{H})$, $\mathcal{F}$ with VC-dimension $d_{vc}(\mathcal{F})$ and*

$$f^*_{S_n} = \underset{f \in \mathcal{F}}{\arg\min}\, R_S(f), g_{S_n}^* = \underset{g \in \mathcal{H}}{\arg\min}\, W_{S_n}\left( g, f^*_{S_n}, \frac{73}{72}\alpha + \frac{\alpha^2}{72} \right),$$

*If $\alpha \in (0, 0.9)$ and:*

$$n \geq \frac{128 \left[ (d_{vc}(\mathcal{H}) + d_{vc}(\mathcal{F})) \log(\frac{1}{\varepsilon}) + \log(\frac{16}{\delta}) \right]}{\epsilon^2 \alpha^2}$$

*We have with probability at least $1 - \delta$:*

 *I)*

$$\mathbb{P}[g_{S_n}^*(x) \geq 0 | g^*(x) \leq 0] = O(\epsilon) \quad [\textit{False Positive}] \tag{34}$$

$$\mathbb{P}[g_{S_n}^*(x) < 0 | g^*(x) \geq 0] \leq O(\epsilon) \quad [\textit{False Negative}] \tag{35}$$

 *II)*

$$\mathbb{P}_{x \sim \mathcal{D}_\alpha}[f^*(x) \neq f_{S_n}^*(x) | g_{S_n}^*(x) \geq 0] = O(\epsilon)$$

*Proof.* I) above can be proved by observing $c = \frac{1 + \frac{1+\alpha}{72}}{2}$ and applying Theorem 3 with Sauer's Lemma to bound the growth function in terms of VC-dimension. Also notice that $c = \frac{1 + \frac{1+\alpha}{72}}{2} \in (\frac{1}{2}, \frac{1+\alpha}{4\alpha})$ given $\alpha < 0.9$. II) can be proved by observing that I) implies $\mathbb{P}[g_{S_n}^*(x) \neq g^*(x)] = O(\varepsilon)$ and by invoking Theorem 1. $\qquad\square$

**Remark 3.** *Let us point out that our proposed selective strategy is different from the consistent selective strategy in (El-Yaniv et al., 2010). Instead of rejecting by looking for consistent output from all hypothesis in the version space, our approach deals with one single reasonably accurate hypothesis (the empirical minimizer). We leverage empirical mistakes made by the predictor in order to learn a selector, aiming to reject (only) the mistakes in a data driven manner. This avoids dealing with the issues found in Theorem 14 in (El-Yaniv et al., 2010), where the selector fails to select any data points.*

**Remark 4.** *In (Cortes et al., 2016), a second hypothesis for the selector is introduced and analyzed, and at the same time, multiple commonly used loss functions are scrutinized and generalization results are provided. The major difference between this work and (Cortes et al., 2016; Geifman & El-Yaniv, 2019) is the motivation pertaining to selective learning. While in (Cortes et al., 2016; Geifman & El-Yaniv, 2019) the selective loss is designed from a coverage ratio perspective, i.e. one wants to trade coverage ratio for a higher precision (selective loss), our approach is designed to distinguish data that is naturally unlearnable and unpredictable. This difference leads to an alternative theoretical result. While the analysis in (Cortes et al., 2016) focuses on selective risk, our theoretical analysis focuses on the quality of the selector in terms of both the false positive and false negative errors with the final goal of rejecting uninformative data with high probability.*

## C Tightness analysis

In this section, we investigate whether it is necessary to have $\tilde{\Omega}\left( \frac{d_{vc}(\mathcal{H}) + d_{vc}(\mathcal{F})}{\alpha^2 \epsilon^2} \right)$ samples to achieve

$$\mathbb{P}_{x \sim \mathcal{D}_\alpha}[f^*(x) \neq f_{S_n}^*(x) | g^*(x) \geq 0] = O(\epsilon)$$

and

$$\max \left\{ \mathbb{P}[g_{S_n}^*(x) \neq g^*(x) | g^*(x) > 0], \mathbb{P}[g_{S_n}^*(x) \neq g^*(x) | g^*(x) \leq 0] \right\} = O(\varepsilon).$$

We will show that there exists a distribution $\mathcal{D}_\alpha$ among the Noisy Generative Process family and hypothesis class $\mathcal{H}, \mathcal{F}$ such that achieving guarantees in Theorem 4 requires $\tilde{\Omega}(\frac{d_{vc}(\mathcal{H}) + d_{vc}(\mathcal{F})}{\alpha^2 \epsilon^2})$ samples, thus making our analysis tight within logarithmic factor. Note that our lower bound is for empirical minimizers of $R(f)$ and $W(g, f, \theta)$; it only illustrates the tightness of our analysis, and thus is not an information theoretic bound for this family of problem - it remains unknown whether strategies other than the empirical minimization of $R(f)$ and $W(g, f, \theta)$ can improve sample complexity to a rate of $\frac{1}{\alpha \varepsilon}$, observing that the informative data is realizable, which usually only requires $O(\frac{1}{\varepsilon})$ samples for an $\epsilon$ statistical error. Our construction leverages the idea from (Ehrenfeucht et al., 1989) about the support of the informative data. Over the support of uninformative data, noise buries the signal of the informative data, thus misleading the empirical minimizer. In our analysis, we will leverage the anti-concentration results of the Chernoff bound, with the proof idea that comes from (Klein & Young, 2015):

**Lemma 6** (tightness of the Chernoff bound). *Let $X$ be the average of $k$ independent, 0/1 random variables (r.v.). For any $\epsilon \in (0, 1/2]$ and $p \in (0, 1/2]$, assuming $\epsilon pk \geq 6, pk \geq 6, \varepsilon \leq \frac{1}{3}$, we have:*

- *If each r.v. is 1 with probability $p$, then*

$$\mathbb{P}[X \leq (1 - \epsilon)p] \geq e^{-6\epsilon^2 pk}.$$

- *If each r.v. is 1 with probability $p$, then*

$$\mathbb{P}[X \geq (1 + \epsilon)p] \geq e^{-6\epsilon^2 pk}.$$

*Proof.* The proof is similar to Lemma 5.2 in (Klein & Young, 2015) with a different choice of parameters. With Stirling's approximation, $i! = \sqrt{2\pi i}(i/e)^i e^\lambda$ with $\lambda \in [1/(12i + 1), 1/12i]$ one can show:

$$\binom{k}{\ell} \geq \frac{1}{e\sqrt{2\pi\ell}} \left(\frac{k}{\ell}\right)^\ell \left(\frac{k}{k - \ell}\right)^{k - \ell} \tag{36}$$

Also note: $\mathbb{P}[X \leq (1 - \epsilon)p] = \sum_{i=0}^{(1-\varepsilon)p} \mathbb{P}[X = \frac{i}{k}]$.

Let $\ell = \lfloor (1 - 2\epsilon)pk \rfloor$. Given the fact that $\varepsilon pk \geq 6$, we have $(1 - 2\epsilon)pk - 1 \leq \ell \leq (1 - 2\epsilon)pk$. It suffices to lower bound $\sum_{i=(1-2\varepsilon)p}^{(1-\varepsilon)p} \mathbb{P}[X = \frac{i}{k}]$ which is at least $\varepsilon pk \mathbb{P}[X = \frac{\ell}{k}]$. From Equation 36 we know that we need to bound $A = \frac{2}{e}\epsilon pk / \sqrt{2\pi\ell}$ and $B = \left(\frac{k}{\ell}\right)^\ell \left(\frac{k}{k-\ell}\right)^{k-\ell} p^\ell (1 - p)^{k-\ell}$. For term $A$, since $\varepsilon pk \geq 6$, $l \leq (1 - 2\varepsilon)pk$ thus we need $pk \geq \frac{9e^{-2\varepsilon}}{\varepsilon}$ to get $\frac{2\varepsilon\sqrt{pk}}{e\sqrt{2\pi(1-2\varepsilon)}} \geq e^{-\varepsilon^2 pk}$. Since $\varepsilon \leq \frac{1}{3}$, it suffices to have $pk \geq 16$. To bound $B$ we need to show:

$$\left(\frac{k}{\ell}\right)^\ell \left(\frac{k}{k - l}\right)^{k-l} p^l (1 - p)^{k-l} \geq e^{-4\varepsilon^2 pk}$$

Since $\left(\frac{k}{\ell}\right)^\ell p^\ell \geq \left(\frac{1}{(1-2\varepsilon)}\right)^l$ and $(1 - p)^{k-l} \left(\frac{k}{k-l}\right)^{k-l} = \left(\frac{(1-p)k}{k(1-p)+1+2\varepsilon pk}\right)^{k-l}$ we have:

$$(1 - 2\varepsilon)^\ell \left(1 + \frac{1 + 2\varepsilon pk}{k(1 - p)}\right)^{k-\ell} \leq e^{-\frac{4\varepsilon^2 p^2 k}{1-p} + 2\varepsilon pk - 2\varepsilon pk + 4\varepsilon^2 pk + 2} \leq e^{5\varepsilon^2 pk}$$

$\square$

We consider the following distribution. Let $d_{vc}(\mathcal{F}) = d$ and $e_j$ be the standard basis with $j$-th coordinate equal to 1 and 0 otherwise. We set $\Omega_U^j = \{+2, -2\} \cdot e_j$, $\Omega_U = \cup_{j=1}^d \Omega_U^j$ and $\Omega_I^j = \{+1, -1\} \cdot e_j$, $\Omega_I = \cup_{j=1}^d \Omega_I^j$. Let $\mathcal{F} = sign(w^\top x)$ and $\mathcal{H} = 2 \prod_{i=1}^n g_j(x) - 1$, where $g_j(x) \in \{\mathbb{1}\{|e_j^\top x| \geq c\} \cup \{0\}, \mathbb{1}\{|e_j^\top x| \leq c\}\}$. By Lemma 3.2.3 in (Blumer et al., 1989) we know $d_{vc}(\mathcal{H}) \leq 6d\log(d)$. We construct $\mathcal{D}_\alpha$ as follows: in $\Omega_U$, each standard basis has $\frac{1-\alpha}{d}$ mass with sign generated via coin flipping (fair). Having generated $x$, $y \in \{+1, -1\}$ is generated via a coin flipping. In $\Omega_I$, we put $\alpha(1 - \varepsilon)$ mass on $e_1$ and $\frac{\alpha\varepsilon}{d-1}$ mass on other standard basis $e_j$, with sign generated via a (fair) coin flipping. Having generated $x$, $y \in \{+1, -1\}$ is generated via $sign(w^{*\top} x)$ where $w^*$ denotes a vector of ones. The above construction allows one to calculate the expected error by counting the number of basis that are not learned correctly, i.e., each $j \in [d]$ s.t. $w_j < 0$ implies an error $\frac{\alpha\varepsilon}{2(d-1)}$ in expectation. Since:

$$\mathbb{E}_{S_n \sim \mathcal{D}_\alpha}[|S_n \cap \Omega_I^j|\,|] = \frac{\alpha\varepsilon n}{d - 1},$$

by Markov inequality we have :

$$\mathbb{P}_{S_n \sim \mathcal{D}_\alpha}[|S_n \cap \Omega_I^j|\,| \geq \frac{6\alpha\varepsilon n}{d - 1}] \leq \frac{1}{6}.$$

Also note that $\mathbb{E}_{x \sim \mathcal{D}_\alpha}[\mathbb{1}\{x \in \Omega_1^j\}\mathbb{1}\{x = -2e_j\}] = \frac{1-\alpha}{2d}$. In order to derive the probability that $w^j < 0$ via empirical minimization, we need to bound the following probability:

$$\mathbb{P}_{x \sim \mathcal{D}_\alpha}\Big[\sum_{x_i \in \Omega_1^j, e_j^\top x_i < 0} y_i - \sum_{x_i \in \Omega_1^j, e_j^\top x_i > 0} y_i \geq \frac{6\alpha\varepsilon n}{d-1}\Big]$$

To do so, we consider the following $n_j$ random variables that are conditioned on $x \in \Omega_1^j$: $z_i = 1$ if $e_j^\top x_i < 0, y_i = 1$ or $e_j^\top x_i \geq 0, y_i = -1$ and $z_i = 0$ otherwise. One can see $2\sum_{i \in n_j} z_i - n_j = \sum_{x_i \in \Omega^j, e_j^\top x_i < 0} y_i - \sum_{x_i \in \Omega^j, e_j^\top x_i > 0} y_i$. Thus :

$$\mathbb{P}_{x \sim \mathcal{D}_\alpha}\Big[\sum_{x_i \in \Omega^j, e_j^\top x_i < 0} y_i - \sum_{x_i \in \Omega^j, e_j^\top x_i > 0} y_i \geq \frac{6\alpha\varepsilon n}{(d-1)}\Big]$$

$$\geq \mathbb{P}_{x \sim \mathcal{D}_\alpha}\Big[\frac{(1-\alpha)n}{d} - \sqrt{\frac{3(1-\alpha)n}{d}} \leq |S_n \cap \Omega_1^j| \leq \frac{3(1-\alpha)n}{d}\Big]$$

$$\cdot \mathbb{P}_{z \sim S_n^j}\Big[\sum_{z_i \in S_n^j} z_i - \frac{n_j}{2} \geq \frac{3\alpha\varepsilon n}{(d-1)}\Big|\frac{(1-\alpha)n}{d} - \sqrt{\frac{3(1-\alpha)n}{d}} \leq n_j \leq \frac{3(1-\alpha)n}{d}\Big]$$

$$\geq \frac{1}{2}\mathbb{P}_{z \sim S_n^j}\Big[\frac{1}{n_j}\sum_{z_i \in S_n^j} z_i - \frac{1}{2} \geq \frac{3\alpha\varepsilon d}{(d-1)(1-\alpha)}\Big|\frac{(1-\alpha)n}{d} - \sqrt{\frac{3(1-\alpha)n}{d}} \leq n_j \leq \frac{3(1-\alpha)n}{d}\Big]$$

$$\geq \frac{1}{2}e^{-18\alpha^2\varepsilon^2 n_j} \geq \frac{1}{2}e^{-\frac{54\alpha^2\varepsilon^2 n}{d}}$$

$$\tag{37}$$

If $n \leq \frac{d}{432\varepsilon^2\alpha^2}$, we have for each $j$, with probability at least $0.40$, $w_j < 0$ due to the empirical minimization strategy. Each $w_j < 0$ contributes $\frac{\alpha\varepsilon}{2(d-1)}$ risk in expectation. With probability $0.2$, $20\%$ of coordinates can not be learned correctly, thus resulting in a risk of at least $0.1\varepsilon\alpha$ in expectation. Next we show that the selector is going to suffer from a poorly learned classifier in terms of its false negative rate. Now lets focus on the case when $\mathbb{E}_{(x,y) \sim \mathcal{D}_\alpha}[f_{S_n}(x) \neq y] \geq \frac{1}{2}(1-\alpha) + 0.1\alpha\varepsilon$ (which happens with probability at least $0.2$), we will show that $\mathbb{E}_{x \sim \mathcal{D}_\alpha}[\mathbb{1}\{g_{S_n}(x) \neq g^*(x)\}|g^*(x) > 0] \geq 0.1\varepsilon$. Let $J_1$ be the set of $j$'s s.t. $w_j < 0$ and $J_2$ for $w_j > 0$. Suppose $\theta > \alpha$, and consider empirical minimization:

$$\min_{g \in \mathcal{H}} W_{S_n}(f_{S_n}^*, g, \theta) \geq \sum_{j=1}^{d} \min_{g \in \mathcal{H}} W_{S_n \cap \Omega^j}(f_{S_n}^*, g, \theta) \tag{38}$$

For $j' \in J_1$, $g \in \arg\min_{g \in \mathcal{H}} W_{S_n \cap \Omega^{j'}}(f_{S_n}^*, g, \theta)$ if $g_{j'}(x) = \mathbb{1}\{|e_{j'}^\top x| \geq c\}, c > 2$ (and for other basis $j$, $g_j$ can be arbitrary). For $j' \in J_2$, $g \in \arg\min_{g \in \mathcal{H}} W_{S_n \cap \Omega^{j'}}(f_{S_n}^*, g, \theta)$ if $g_{j'}(x) = \mathbb{1}\{|e_{j'}^\top x| \leq c\}, c \in [1, 2)$. Now consider $g_{S_n}^* = 2\prod_{j=1}^{d} g_j(x) - 1, g_j(x) = \mathbb{1}\{|e_j^\top x| \geq c\}$ with $c > 2$ if $j \in J_1$ and $g_j(x) = \mathbb{1}\{|e_j^\top x| \leq c\}, c \in [1, 2)$ if $j \in J_2$. One can see that such $g_{S_n}^*$ is in the intersection of set of minimizers of $W_{S_n \cap \Omega^j}(f_{S_n}^*, g, \theta)$ thus attains the minimum. The total mass of $x$ s.t. $g_{S_n}^*(x) \neq g^*(x)$ is at least $\frac{|J_1|\alpha\varepsilon}{d-1}$ since for all $x \in \Omega^j$ with $j \in \mathcal{J}_1, g_{S_n}^*(x) \neq g^*(x)$. The event $\mathcal{E} : \mathbb{E}_{(x,y) \sim \mathcal{D}_\alpha}[f_{S_n}(x) \neq f^*(x)] \geq \frac{1}{2}(1-\alpha) + 0.1\alpha\varepsilon$ happens with probability at least $0.2$ which implies that $|\mathcal{J}_1| \geq 0.2d$. Thus we have following lower bound on the False Negative:

$$\mathbb{P}_{x \sim \mathcal{D}_\alpha}[g_{S_n}^*(x) \neq g^*(x)|g^*(x) > 0] \geq 0.1\varepsilon.$$

## D   SUPPLEMENTAL MATERIAL FOR EMPIRICAL STUDY

**Extension to Multi-class.** Our method extends to multi-class setting naturally. In the case of $K$-class classification, we predictors $f(x) = f(x)_{1:K} : \mathcal{X} \rightarrow \Delta^K$ where $\Delta^K$ is the $K$-simplex. In multi-class scenario, our selector loss remains the same. Meanwhile, we use multi-class cross entropy loss to train classifier. The pseudo-informative label becomes $z_i = \mathbb{1}\{\arg\max_{k \in [K]} f(x_i)_k = y_i\}$.

**Experiment setting with Unknown $\alpha$ (More Details).** For MNIST/Fashion-MNIST experiment, the backbone model we use for each candidate is TinyCNN, which is a CNN with two consecutive convolution layers and 3 consecutive fully connected layers. This architecture provides enough model capacity for an MNIST-type dataset. We use 196 as the batch size. We use Adam as the optimizer with learning rate 1e-3 and weight decay 1e-4. We train with each algorithm for 162 epochs. We use the default hyper-parameters for every method (i.e., internal selective learning-specific defaults, as reported in (Geifman & El-Yaniv, 2019; Liu et al., 2019), and $\beta = \frac{\theta(1-\alpha)}{\alpha(1-\theta)}$ for our algorithm). It simulates a practical scenario where hyper-parameter optimization is impossible, since the ground truth regarding which datapoints are actually learnable is never revealed. For the SVHN experiment, the backbone model we use is ResNet18 (He et al., 2016b) for every candidate. We use 128 as the batch size. We use Adam as the optimizer with learning rate 1e-3 and weight decay 1e-4. We train each algorithm for 162 epochs.

**Weight of Informative data.** From Figure 3, we can see that our algorithm can pick out almost all informative data by the end of training for all informative/uninformative ratios even without knowing the ground-truth $\alpha$.

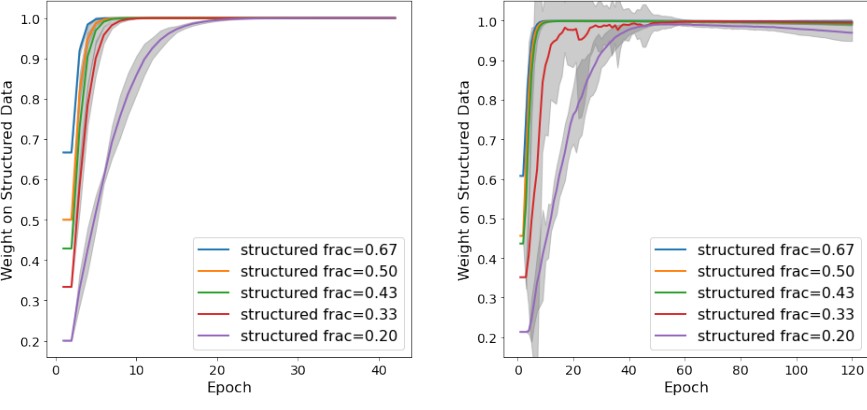

(a) MNIST. Weight on Informative Data     (b) SVHN. Weight on Informative Data

Figure 3: Weight of informative data as a function of training epoch and the ratio of informative data. The $y$ axis is the percentage of weight put on the iformative data, i.e $\frac{\sum_{i:x_i \in \Omega_I} \gamma_i}{\sum_i^n \gamma_i}$ in the notation of Algorithm 1. Each experiment is repeated 3 times, and 3 standard deviations are presented as the shaded gray area.

**Experiments with Known $\alpha$.** We also test the performance of each method in the idealized case. In this experiment, we give the groud-truth $\alpha$ to each baseline and perform an exhaustive HPO (Hyper Parameter Optimization). We use the same backbone, optimizer and the number of training epochs as before. We run a random hyper-parameter search 500 times (selective loss weight and parameter $\lambda$ for SelectiveNet and parameter $o$ in DeepGambler) and record the best performance of each baseline. In this setting, our method still outperform baselines for most levels of noise ratios. In this experiment, we use two stage training for our algorithm. We first run 120 epochs to finish 1st round data selection and then run 42 epochs to train a classifier only with these selected data. The performance is measured with this refined the classifier.

**Experiment with Low Ratio of Uninformative Data.** We also conduct experiments with datasets that contain minority stochastic data. This experiment is conducted under both practical (unknown $\alpha$) and idealized scenario (known $\alpha$). In practical setting (unknown $\alpha$), our proposed algorithm still wins by a large margin. In the case when $\alpha$ is known, DeepGambler outperforms other methods in

most cases. We hypothesize that when noisy data minority, our method doesn't get enough samples to fully learn the pattern of uninformative data.

## D.1 EXPERIMENT WITH KNOWN $\alpha$

Table 5: (MNIST - Full Data Setting - Known $\alpha$) Results on a synthetic dataset consisting of uninformative MNIST data and informative Fashion-MNIST data using the entire MNIST.

| Uninformative Data Num. | Informative Data Num. | Criterion | Confidence | SelectiveNet | DeepGambler | Ours |
|---|---|---|---|---|---|---|
| 60000 | 15000 | SR(%) | $10.00 \pm 0.32$ | $10.11 \pm 0.42$ | $9.77 \pm 0.51$ | **9.18 ± 0.49** |
| | | Precision | $0.99 \pm 0.00$ | $1.00 \pm 0.00$ | $1.00 \pm 0.00$ | $1.00 \pm 0.00$ |
| | | Recall | $0.99 \pm 0.00$ | $1.00 \pm 0.00$ | $1.00 \pm 0.00$ | $1.00 \pm 0.00$ |
| 60000 | 30000 | SR(%) | $9.41 \pm 0.20$ | $9.91 \pm 0.48$ | **9.39 ± 0.26** | **9.03 ± 0.73** |
| | | Precision | $0.99 \pm 0.00$ | $1.00 \pm 0.00$ | $1.00 \pm 0.00$ | $1.00 \pm 0.00$ |
| | | Recall | $0.99 \pm 0.00$ | $1.00 \pm 0.00$ | $1.00 \pm 0.00$ | $1.00 \pm 0.00$ |
| 60000 | 45000 | SR(%) | $8.63 \pm 0.23$ | $9.39 \pm 0.35$ | $9.13 \pm 0.51$ | **8.58 ± 0.26** |
| | | Precision | $1.00 \pm 0.00$ | $1.00 \pm 0.00$ | $1.00 \pm 0.00$ | $1.00 \pm 0.00$ |
| | | Recall | $0.99 \pm 0.00$ | $1.00 \pm 0.00$ | $1.00 \pm 0.00$ | $1.00 \pm 0.00$ |
| 60000 | 60000 | SR(%) | $8.11 \pm 0.08$ | $8.21 \pm 0.12$ | $8.16 \pm 0.05$ | $8.04 \pm 0.49$ |
| | | Precision | $1.00 \pm 0.00$ | $1.00 \pm 0.00$ | $1.00 \pm 0.00$ | $1.00 \pm 0.00$ |
| | | Recall | $0.99 \pm 0.00$ | $1.00 \pm 0.00$ | $1.00 \pm 0.00$ | $1.00 \pm 0.00$ |

Table 6: (MNIST - Partial Data Setting - Known $\alpha$) Results on a synthetic dataset consisting of uninformative MNIST data and informative Fashion-MNIST data using 25% of MNIST.

| Uninformative Data Num. | Informative Data Num. | Criterion | Confidence | SelectiveNet | DeepGambler | Ours |
|---|---|---|---|---|---|---|
| 15000 | 3750 | SR(%) | $16.71 \pm 0.31$ | $13.50 \pm 0.30$ | $14.11 \pm 5.65$ | **11.38 ± 0.49** |
| | | Precision | $0.94 \pm 0.02$ | $1.00 \pm 0.00$ | $1.00 \pm 0.00$ | $1.00 \pm 0.00$ |
| | | Recall | $0.90 \pm 0.04$ | $1.00 \pm 0.00$ | $1.00 \pm 0.00$ | $1.00 \pm 0.00$ |
| 15000 | 7500 | SR(%) | $13.93 \pm 0.40$ | $13.34 \pm 1.23$ | **11.20 ± 0.22** | **11.29 ± 0.44** |
| | | Precision | $0.96 \pm 0.01$ | $0.99 \pm 0.01$ | $1.00 \pm 0.00$ | $1.00 \pm 0.00$ |
| | | Recall | $0.95 \pm 0.02$ | $0.93 \pm 0.16$ | $1.00 \pm 0.00$ | $1.00 \pm 0.00$ |
| 15000 | 11250 | SR(%) | $12.29 \pm 0.31$ | $11.45 \pm 0.37$ | **10.61 ± 0.12** | **10.42 ± 0.48** |
| | | Precision | $0.97 \pm 0.01$ | $1.00 \pm 0.00$ | $1.00 \pm 0.00$ | $0.99 \pm 0.00$ |
| | | Recall | $0.95 \pm 0.03$ | $1.00 \pm 0.00$ | $1.00 \pm 0.00$ | $1.00 \pm 0.00$ |
| 15000 | 15000 | SR(%) | $11.76 \pm 0.20$ | $11.24 \pm 0.35$ | **10.12 ± 0.28** | **9.97 ± 0.33** |
| | | Precision | $0.97 \pm 0.01$ | $1.00 \pm 0.00$ | $1.00 \pm 0.00$ | $0.99 \pm 0.01$ |
| | | Recall | $0.95 \pm 0.02$ | $1.00 \pm 0.00$ | $1.00 \pm 0.00$ | $1.00 \pm 0.00$ |

Table 7: (SVHN - Full Data Setting - Known $\alpha$) Results on a synthetic dataset consisting of uninformative SVHN and informative SVHN using the entire uninformative data.

| Uninformative Data Num. | Informative Data Num. | Criterion | Confidence | SelectiveNet | DeepGambler | Ours |
|---|---|---|---|---|---|---|
| 33800 | 9200 | SR(%) | $6.45 \pm 0.93$ | $58.06 \pm 35.91$ | $4.58 \pm 0.61$ | **4.48 ± 0.43** |
| | | Precision | $0.93 \pm 0.01$ | $0.41 \pm 0.36$ | $0.96 \pm 0.00$ | $0.96 \pm 0.00$ |
| | | Recall | $0.89 \pm 0.01$ | $0.37 \pm 0.34$ | $0.86 \pm 0.02$ | $0.86 \pm 0.01$ |
| 33800 | 18300 | SR(%) | $4.30 \pm 0.31$ | $80.42 \pm 1.55$ | $3.08 \pm 0.15$ | **2.91 ± 0.21** |
| | | Precision | $0.96 \pm 0.00$ | $0.35 \pm 0.00$ | $0.97 \pm 0.00$ | $0.97 \pm 0.00$ |
| | | Recall | $0.91 \pm 0.01$ | $0.38 \pm 0.05$ | $0.90 \pm 0.01$ | $0.88 \pm 0.01$ |
| 33800 | 26200 | SR(%) | $4.49 \pm 0.68$ | $5.22 \pm 0.90$ | $3.86 \pm 0.11$ | **3.65 ± 0.17** |
| | | Precision | $0.96 \pm 0.68$ | $0.95 \pm 0.01$ | $0.97 \pm 0.00$ | $0.96 \pm 0.00$ |
| | | Recall | $0.94 \pm 0.01$ | $0.92 \pm 0.03$ | $0.94 \pm 0.00$ | $0.93 \pm 0.01$ |
| 33800 | 28400 | SR(%) | $10.87 \pm 0.68$ | $8.27 \pm 5.03$ | $7.65 \pm 0.47$ | **6.39 ± 0.76** |
| | | Precision | $0.89 \pm 0.00$ | $0.92 \pm 0.05$ | $0.92 \pm 0.01$ | $0.93 \pm 0.01$ |
| | | Recall | $0.98 \pm 0.00$ | $0.95 \pm 0.04$ | $0.97 \pm 0.01$ | $0.95 \pm 0.01$ |

Table 8: (SVHN - Partial Data Setting - Known $\alpha$) Results on a synthetic dataset consisting of uninformative SVHN and informative SVHN using 25% uninformative data.

| Uninformative Data Num. | Informative Data Num. | Criterion | Confidence | SelectiveNet | DeepGambler | Ours |
|---|---|---|---|---|---|---|
| 9200 | 2285 | SR(%) | $30.13 \pm 34.94$ | $55.91 \pm 29.49$ | **7.65 ± 0.48** | $11.74 \pm 2.32$ |
| | | Precision | $0.70 \pm 0.35$ | $0.19 \pm 0.10$ | $0.93 \pm 0.01$ | $0.90 \pm 0.02$ |
| | | Recall | $0.67 \pm 0.34$ | $0.16 \pm 0.13$ | $0.78 \pm 0.02$ | $0.80 \pm 0.06$ |
| 9200 | 4600 | SR(%) | $10.41 \pm 0.26$ | $36.46 \pm 28.65$ | $9.70 \pm 1.05$ | **8.12 ± 0.87** |
| | | Precision | $0.91 \pm 0.00$ | $0.68 \pm 0.24$ | $0.90 \pm 0.03$ | $0.93 \pm 0.01$ |
| | | Recall | $0.87 \pm 0.01$ | $0.69 \pm 0.24$ | $0.79 \pm 0.04$ | $0.78 \pm 0.08$ |
| 9200 | 6900 | SR(%) | $8.51 \pm 0.85$ | $34.97 \pm 30.15$ | **7.49 ± 0.47** | **7.67 ± 0.63** |
| | | Precision | $0.92 \pm 0.02$ | $0.73 \pm 0.21$ | $0.94 \pm 0.01$ | $0.92 \pm 0.02$ |
| | | Recall | $0.91 \pm 0.02$ | $0.74 \pm 0.22$ | $0.83 \pm 0.04$ | $0.86 \pm 0.01$ |
| 9200 | 9200 | SR(%) | $9.06 \pm 0.82$ | $13.66 \pm 1.39$ | $7.88 \pm 0.29$ | **7.57 ± 0.51** |
| | | Precision | $0.91 \pm 0.01$ | $0.88 \pm 0.02$ | $0.93 \pm 0.01$ | $0.92 \pm 0.01$ |
| | | Recall | $0.94 \pm 0.01$ | $0.92 \pm 0.01$ | $0.88 \pm 0.03$ | $0.91 \pm 0.01$ |

Table 9: Results on low noise level for both MNIST and SVHN - Unknown $\alpha$. We evaluate the methods over different fractions of informative data.

| Data Set | Uninformative Data Num. | Informative Data Num. | Criterion | Standard | SelectiveNet | DeepGambler | Ours |
|---|---|---|---|---|---|---|---|
| MNIST | 30000 | 60000 | SR(%) | $21.59 \pm 0.33$ | $14.97 \pm 0.00$ | $14.52 \pm 0.76$ | $\mathbf{12.97 \pm 2.84}$ |
| | | | Precision | $0.86 \pm 0.00$ | $0.93 \pm 0.00$ | $0.94 \pm 0.00$ | $1.00 \pm 0.00$ |
| | | | Recall | $0.98 \pm 0.00$ | $1.00 \pm 0.00$ | $1.00 \pm 0.00$ | $0.98 \pm 0.02$ |
| MNIST | 7500 | 15000 | SR(%) | $34.73 \pm 0.87$ | $36.17 \pm 0.00$ | $34.20 \pm 0.01$ | $\mathbf{18.72 \pm 6.31}$ |
| | | | Precision | $0.72 \pm 0.01$ | $0.72 \pm 0.01$ | $0.74 \pm 0.01$ | $0.98 \pm 0.00$ |
| | | | Recall | $0.93 \pm 0.01$ | $1.00 \pm 0.00$ | $1.00 \pm 0.00$ | $0.90 \pm 0.07$ |
| SVHN | 18300 | 28400 | SR(%) | $30.59 \pm 0.55$ | $8.94 \pm 0.00$ | $7.32 \pm 0.01$ | $\mathbf{4.69 \pm 0.00}$ |
| | | | Precision | $0.71 \pm 0.01$ | $0.92 \pm 0.01$ | $0.93 \pm 0.01$ | $0.96 \pm 0.00$ |
| | | | Recall | $0.99 \pm 0.00$ | $0.99 \pm 0.00$ | $0.99 \pm 0.00$ | $0.92 \pm 0.01$ |
| SVHN | 2300 | 9200 | SR(%) | $36.69 \pm 0.85$ | $16.04 \pm 0.01$ | $17.48 \pm 0.02$ | $\mathbf{6.08 \pm 0.01}$ |
| | | | Precision | $0.65 \pm 0.01$ | $0.86 \pm 0.02$ | $0.85 \pm 0.02$ | $0.95 \pm 0.01$ |
| | | | Recall | $0.97 \pm 0.00$ | $0.98 \pm 0.01$ | $0.98 \pm 0.00$ | $0.90 \pm 0.01$ |

Table 10: Results on low noise level for both MNIST and SVHN - Known $\alpha$. We evaluate the methods over different fractions of informative data.

| Data Set | Uninformative Data Num. | Informative Data Num. | Criterion | Standard | SelectiveNet | DeepGambler | Ours |
|---|---|---|---|---|---|---|---|
| MNIST | 30000 | 60000 | SR(%) | $8.19 \pm 0.32$ | $8.17 \pm 0.12$ | $8.09 \pm 0.11$ | $\mathbf{7.97 \pm 0.13}$ |
| | | | Precision | $1.00 \pm 0.00$ | $1.00 \pm 0.00$ | $1.00 \pm 0.00$ | $1.00 \pm 0.00$ |
| | | | Recall | $0.99 \pm 0.00$ | $1.00 \pm 0.00$ | $1.00 \pm 0.00$ | $1.00 \pm 0.00$ |
| MNIST | 7500 | 15000 | SR(%) | $11.65 \pm 0.38$ | $11.47 \pm 0.21$ | $\mathbf{9.75 \pm 0.63}$ | $10.37 \pm 0.13$ |
| | | | Precision | $0.93 \pm 0.00$ | $0.99 \pm 0.00$ | $1.00 \pm 0.01$ | $1.00 \pm 0.00$ |
| | | | Recall | $0.87 \pm 0.01$ | $1.00 \pm 0.00$ | $0.86 \pm 0.28$ | $1.00 \pm 0.00$ |
| SVHN | 18300 | 28400 | SR(%) | $11.81 \pm 0.29$ | $12.18 \pm 0.67$ | $\mathbf{6.92 \pm 0.43}$ | $9.39 \pm 0.92$ |
| | | | Precision | $0.89 \pm 0.00$ | $0.89 \pm 0.01$ | $0.93 \pm 0.00$ | $0.89 \pm 0.01$ |
| | | | Recall | $0.99 \pm 0.00$ | $0.98 \pm 0.00$ | $0.98 \pm 0.00$ | $0.98 \pm 0.01$ |
| SVHN | 2300 | 9200 | SR(%) | $8.49 \pm 0.13$ | $11.96 \pm 1.18$ | $\mathbf{6.81 \pm 0.60}$ | $8.43 \pm 1.41$ |
| | | | Precision | $0.92 \pm 0.01$ | $0.89 \pm 0.01$ | $0.95 \pm 0.01$ | $0.92 \pm 0.02$ |
| | | | Recall | $0.95 \pm 0.01$ | $0.95 \pm 0.01$ | $0.90 \pm 0.01$ | $0.95 \pm 0.01$ |

## D.2 Experiment with Low Ratio of Uninformative Data
## D.3 Implementation Detail

For all our experiments, we are using NVIDIA Tesla V100 GPUs, and we use AWS Sagemaker service for our HPO processes. We repeat each experiment 5 times with random seed 77, 78, 79, 80, 81.

For MNIST/Fashion-MNIST experiment where $\alpha$ is unknown, the training and testing data set split is listed in Table 11. The backbone we use is TinyCNN (in section 4). We use 196 as our batch size and train for 162 epochs for all listed methods. We use Adam as the optimizer with learning rate 1e-3 and weight decay 1e-4. We shrink our learning rate at epoch 45 and 90 by half each time.

For SVHN experiment where $\alpha$ is unknown, the training and testing data set split is listed in Table 11. The backbone we use is ResNet18. We use 128 as our batch size and train for 162 epochs for all listed methods. We use Adam as the optimizer with learning rate 1e-3 and weight decay 1e-4. We shrink our learning rate at epoch 45 and 90 by half each time. For both experiments where $\alpha$ is known

Table 11: Experiment Data Splitting.

| Data Set | Uninformative Training Data Num. | Informative Training Data Num. | Uninformative Testing Data Num. | Informative Training Data Num. |
|---|---|---|---|---|
| MNIST | 60000 | 60000 | 10000 | 10000 |
| | 60000 | 45000 | 10000 | 7500 |
| | 60000 | 30000 | 10000 | 5000 |
| | 60000 | 15000 | 10000 | 2500 |
| | 30000 | 60000 | 5000 | 10000 |
| | 15000 | 15000 | 10000 | 10000 |
| | 15000 | 11250 | 10000 | 7500 |
| | 15000 | 7500 | 10000 | 5000 |
| | 15000 | 3750 | 10000 | 2500 |
| | 7500 | 15000 | 5000 | 10000 |
| SVHN | 33800 | 28400 | 11900 | 9800 |
| | 33800 | 26000 | 11900 | 9100 |
| | 33800 | 18300 | 11900 | 6500 |
| | 33800 | 9200 | 11900 | 3300 |
| | 18300 | 28400 | 6500 | 9800 |
| | 9200 | 9200 | 11900 | 9800 |
| | 9200 | 6900 | 11900 | 9100 |
| | 9200 | 4600 | 11900 | 6500 |
| | 9200 | 2300 | 11900 | 3300 |
| | 4600 | 9200 | 6500 | 9800 |

and HPO is enabled, we maintain the same backbone and optimizer. The hyper-parameter setting is documented in the code. The code will be open sourced together with the publication of the paper.

# E    EXTENTION OF THEOREM 4 IN THE CASE WHERE ONLY AN UPPER BOUND ON $\alpha$ IS GIVEN

**Theorem 5.** *Under Assumption 1, given set of samples* $S_n = \{(x_1, y_1), ..., (x_n, y_n)\}$ *drawn i.i.d. from the Noisy Generative Process with* $\alpha \in (0, 1 - \xi], \xi \geq 0.2$ , *hypothesis class* $\mathcal{H}$ *with VC-dimension* $d_{vc}(\mathcal{H})$, $\mathcal{F}$ *with VC-dimension* $d_{vc}(\mathcal{F})$ *and* $f^*_{S_n} = \arg\min_{f \in \mathcal{F}} R_S(f), g_{S_n}^* = \arg\min_{g \in \mathcal{H}} W_{S_n}\left(g, f^*_{S_n}, \theta\right)$. *If* $\theta \in \left(\frac{73\alpha}{72}, min\{1 + \frac{\xi}{2(1-\xi)}, 10\}\alpha\right)$ *and* $n \geq \frac{3200\left[(d_{vc}(\mathcal{H}) + d_{vc}(\mathcal{F}))\log(\frac{1}{\epsilon}) + \log(\frac{16}{\delta})\right]}{\epsilon^2 \alpha^2}$. *We have with probability at least* $1 - \delta$:

*I)*

$$\mathbb{P}[g^*_{S_n}(x) \geq 0 | g^*(x) \leq 0] = O(\epsilon) \quad [\textit{False Positive}]$$
$$\mathbb{P}[g^*_{S_n}(x) < 0 | g^*(x) \geq 0] \leq O(\epsilon) \quad [\textit{False Negative}]$$

(39)

*II)*

$$\mathbb{P}_{x \sim \mathcal{D}_\alpha}[f^*(x) \neq f^*_{S_n}(x) | g^*_{S_n}(x) \geq 0] = O(\epsilon)$$

*III) Further more, to obtain* $g_{S_n}^* = \arg\min_{g \in \mathcal{H}} W_{S_n}\left(g, f^*_{S_n}, \theta\right)$ *it suffices to minimize following loss with* $\beta \in (1 + \frac{1}{73\xi - 1}, min\{1 + \frac{\xi}{2(1-\xi)}, 10\})$:

$$Loss(g; f^*_{S_n}, S_n, \beta) = \sum_{i=1}^{n}\left\{\beta\mathbb{1}\{f^*_{S_n}(x_i) \neq y_i\}\mathbb{1}\{g(x_i) > 0\} + \mathbb{1}\{f^*_{S_n}(x_i) = y_i\}\mathbb{1}\{g(x_i) \leq 0\}\right\}$$

*Proof.* I) above can be proved by picking $c \in (\frac{1}{2} + \frac{1}{144}, \frac{1}{2} + min\{\frac{9}{2}, \frac{\xi}{4(1-\xi)}\})$ in Theorem 3. Note this range is valid since $\alpha \leq 1 - \xi$, we have

$$\alpha \leq 1 - \xi \iff \underset{\leq 1}{\underbrace{\frac{\alpha}{1 - \xi}}} \leq \underset{\geq 1}{\underbrace{\frac{1 - \alpha}{\xi}}}$$
$$\iff \frac{\xi}{1 - \xi} \leq \frac{1 - \alpha}{\alpha}$$
$$\iff \frac{\xi}{2(1 - \xi)} \leq \frac{1 + \alpha}{2\alpha} - 1$$
$$\iff \frac{\xi}{4(1 - \xi)} + \frac{1}{2} \leq \frac{1 + \alpha}{4\alpha}$$

(40)

By a procedure similar to Theorem 4 (using the Sauer's Lemma) we have I). Note $c \in (\frac{1}{2} + \frac{1}{144}, \frac{1}{2} + min\{1, \frac{\xi}{4(1-\xi)}\}) \iff \theta \in \left(\frac{73\alpha}{72}, min\{1 + \frac{\xi}{2(1-\xi)}, 2\}\alpha\right)$. II) can be proved by observing that I) implies $\mathbb{P}[g^*_{S_n}(x) \neq g^*(x)] = O(\varepsilon)$ and by invoking Theorem 1. Next we prove III). It is easy to verify that the minimizer of $W_{S_n}(g, f^*_{S_n})$ is equivalent to minimizer of $Loss(g; f^*_{S_n}, S_n, \beta)$ (by replacing $\beta = \frac{\theta(1-\alpha)}{\alpha(1-\theta)}$). It suffices to analyze that $\beta \in (1 + \frac{1}{73\xi - 1}, min\{1 + \frac{\xi}{2(1-\xi)}, 10\})$ implies that $\theta \in \left(\frac{73\alpha}{72}, min\{1 + \frac{\xi}{2(1-\xi)}, 10\}\alpha\right)$. First note that $\frac{x}{1-x}$ is monotone increasing in $(0, 1)$, it suffices to compute image of $\frac{\theta(1-\alpha)}{\alpha(1-\theta)}$ and check if $\beta \in (1 + \frac{1}{73\xi - 1}, min\{1 + \frac{\xi}{2(1-\xi)}, 10\})$ stays in such range. It can be easily verified that $\alpha \leq 1 - \xi$ implies the lower bound $1 + \frac{1}{73\xi - 1}$ and $\alpha > 0$ implies the upper bound $1 + \frac{\xi}{2(1-\xi)}$.

$\square$

# F    ABLATION STUDY ON HYPERPARAMETERS

We provide an ablation study where we vary the hyper-parameter in each baseline. For SelectiveNet, we first fix $\lambda$ to be 32 (default setting that is recommended by the author in the original paper) and then we vary $a$ from 0.1 to 0.7. Then we fix $a$ to be 0.5 (default setting) and then we vary $\lambda$

from 1 to 66. For DeepGambler, we vary $o$ from 1 to 7. Finally, for our algorithm, we increase the hyper-parameter $\beta$ 4 to 10. We can see that the performance of all baselines are quite sensitive to the choice of hyper-parameter and will experience some large fluctuation. On the contrast, our algorithm is stable against the choice of hyper-parameter. This empirical observation supports that there exists a wide range of $\beta$ so that one can control the FNR and FPR if $\alpha$ is bounded away from 1 as it is implied in Theorem 5. Furthermore, in all scenario, our algorithm's performance is better than these two baselines in two sense. On one hand, our selector has better precision such that we can recover almost all informative data while the other two cannot. These two baseline tend to to select the whole data set indistinguishably (low precision and high recall). On the other hand, these baselines show consistently deteriorated risk performance than ours because of their selector fails to pick informative data.

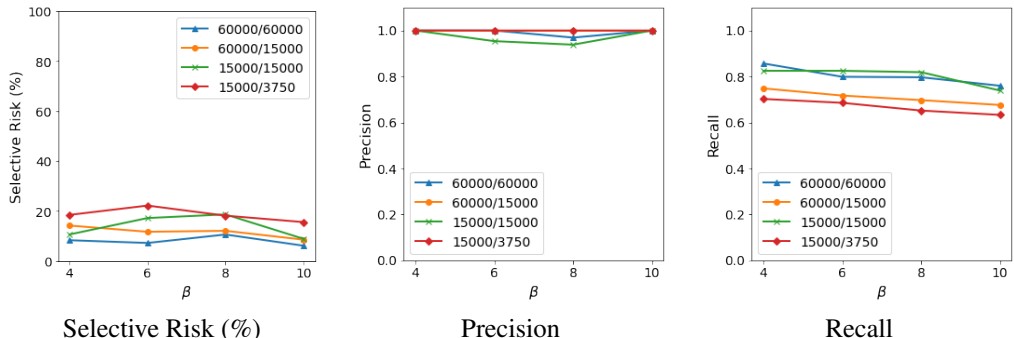

Figure 4: Ablation Study on Hyper-parameter $\beta$ - Our Method.

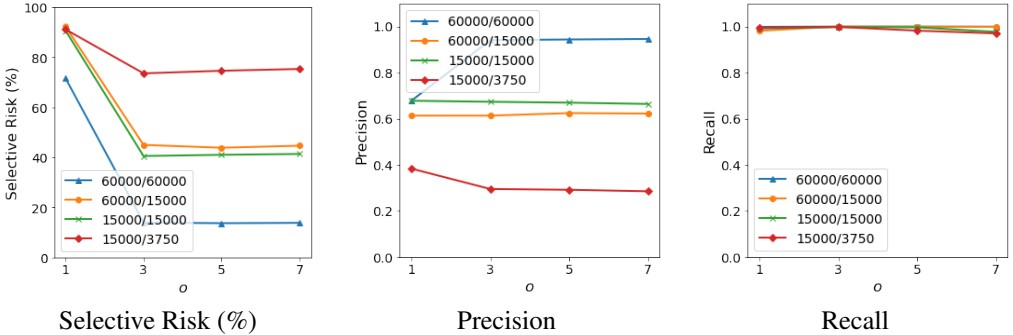

Figure 5: Ablation Study on Hyper-parameter $o$ - DeepGambler.

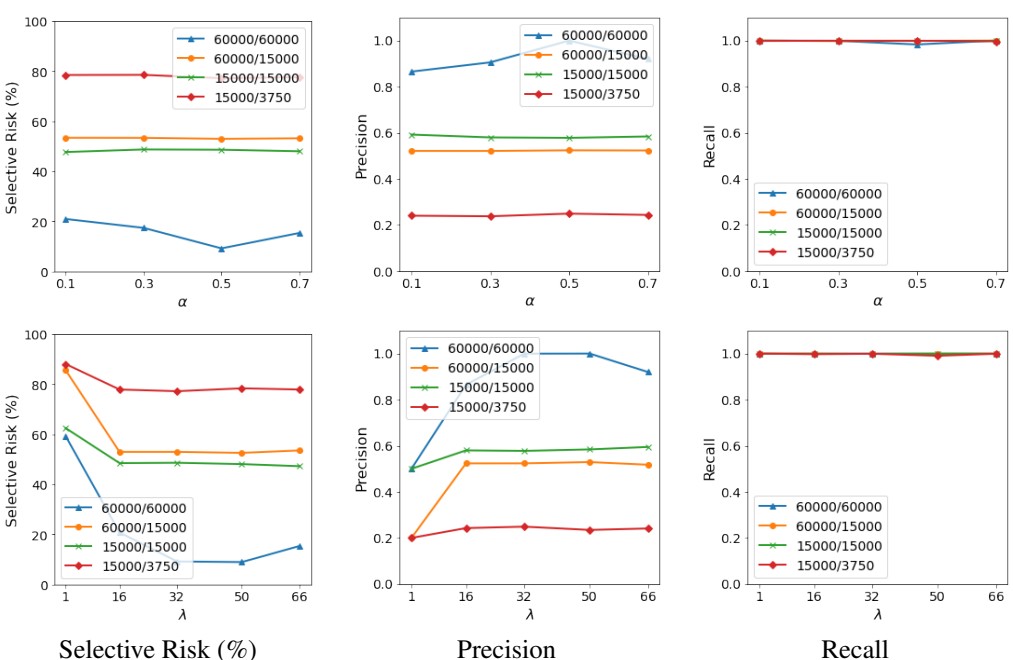

Figure 6: Ablation Study on Hyper-parameter $a$ and $\lambda$ - SelectiveNet.

