# OpenReview forum: "Learning to Abstain in the Presence of Uninformative Data"
_ICLR.cc/2022/Conference — ICLR 2022 Submitted_

### Official Review · Reviewer_zieu · 2021-11-02

**Correctness:** 4
**Technical Novelty And Significance:** 3
**Empirical Novelty And Significance:** 2
**Recommendation:** 6
**Confidence:** 2

**Main Review:**

The paper is well-written, clearly organized and easy to follow. The problem of identifying uninformative data is interesting and very relevant in modern machine learning. The theory of the paper---though highly stylized and is based on $0-1$ loss that cannot be optimized efficiently---is revealing and also a good starting point. However, this is also the main weakness of the paper, the algorithm analyzed is based on $0-1$ loss that cannot be solved efficiently, while the heuristic algorithm implemented is based on the cross-entropy surrogate.

Some detailed comments are listed below:

1. In the setup that the uninformative part consists of pure noise while the informative part is noiseless in Eq. (2), which seems a bit unrealistic. Can the authors explain why in the informative part of the data cannot be random? (say $\mathsf{sign}(P(y=1|x) - 1/2) = f^\star(x)$ under some Mammen-Tsybakov condition/margin condition)
2. The empirical studies are not exactly supported by the theory as they are essentially different algorithms.
3. Can the authors specify what are the hypothesis classes in the experimental setup?



**Summary Of The Paper:**

The authors analyze selective learning when part of the data is pure noise and uninformative. The paper presents an algorithm by a two-phase optimization procedure and provide PAC bounds for both the ground truth predictor $f^\star$ and the selector $g^\star$.

**Summary Of The Review:**

The paper studies identifying uninformative data in datasets which is a very relevant problem for the machine learning community. The algorithm analyzed in the paper is supported by interesting theoretical results and can be revealing for more realistic algorithms.

---

> ### Author Response · Authors · 2021-11-19
> **Response to Reviewer zieu**
>
> Thank you for your constructive feedback and insightful questions. We address your questions/concerns as follows:
>
> ***Q1. In the setup, the uninformative part consists of pure noise while the informative part is noiseless in Eq. (2), which seems a bit unrealistic. Can the authors explain why the informative part of the data cannot be random? (say $sign(P(y=1|x)−\frac{1}{2})=f^{\ast}(x)$ under some Mammen-Tsybakov condition/margin condition)***
>
> ___Answer___:  This is a good question! We believe our method can be extended to random informative data with Massart Noise or Mammen-Tsybakov condition. While the sample complexity of learning a good classifier only requires little extra work with a dependence on the margin,  learning selector will require more involved analysis. It will need a careful choice on the hyperparameter since the selector may incur false positive error in the informative region due to the label noise.  This is an interesting direction for future research.
>
>
> ***Q2. The empirical studies are not exactly supported by the theory as they are essentially different algorithms.***
>
> ___Answer___: Our practical algorithm is inspired by our theoretical analysis. To extend the theoretical results to this practical algorithm, we need 1) more involved analysis for cross entropy loss which is not uniformly bounded, 2) analyzing the iterative convergence property of the algorithm. We look forward to such extension  in future work.
>
>
> ***Q3. Can the authors specify what are the hypothesis classes in the experimental setup?***
>
> ___Answer___: In the experimental setup the hypotheses are deep neural networks with certain architecture (included in the appendix). We will make this clear in the experiment section.

---

### Official Review · Reviewer_QnTY · 2021-11-02

**Correctness:** 4
**Technical Novelty And Significance:** 4
**Empirical Novelty And Significance:** 3
**Recommendation:** 6
**Confidence:** 2

**Main Review:**

The main contribution of the paper is a new model for prediction with abstention, which postulates that samples in one region of the feature space can be predicted well, while in the rest of the space entirely unpredictable.  Under this model, the authors derive an oracle-based algorithm which can provably recover both the informative-partition (selector) and a good model on it (predictor).  For practical settings one can not always assume exact minimization of the empirical risk, so the authors propose an iterative algorithm which alternates learning the selector and the predictor, using something akin to multiplicative updates. Under the assumed generative models it seems to outperform baselines, sometimes by a wide margin, especially in a setting where the fraction of informative samples is small.

The paper mostly reads well, has detailed theoretical analysis (which I only had a chance to skim), and presents experimental evidence supporting the claims.

My main criticism of the paper is:
1) To me the model seems entirely artificial, and I am not aware of any realistic settings where the data can be partitioned into a region with great classification, and an uninformative region. For high-noise data in my experience it's much more common that there is a fairly weak signal over the entire feature space, with no pockets of high predictability. I'd be very interested to hear evidence on the contrary.  The examples are very stylized, and likely heavily leverage the fact that they match the assumed generative model. Showing how this translates to a more realistic example would have been helpful -- as the analysis may have a chance to be suggestive for more practical settings as well.

2) I am surprised that using methods which allow uncertainty quantification -- e.g. maybe bootstrap resampling, or say Gaussian processes would have a hard time in this setting. I would have expected that it's not that hard to learn to distinguish MNIST from fashion MNIST, and to assign tight confidences around the region of MNIST examples, while giving large error bars to fashion-MNIST. Is the problem that the "Model prediction confidence" gives out incorrect (i.e. uncalibrated confidences?).  For DNNs and many other learners this overconfidence is a well-known problem (e.g. there's a nice paper "On Calibration of Modern Neural Networks" from Killian Weinberger's group,  and also well known solutions, for example from Charler Elkan and Bianca Zadrozny, "Obtaining calibrated probability estimates for classifiers".

3) If instead of trying to identify the "predictable region" -- you try to apply / modify the method to attack the original selective prediction -- is it competitive with existing approaches? My suspicion is that the approach is quite dependent on having data with a well-separated true informative/uninformative partition -- and going after support recovery.

4) How do you incorporate the complexity of the decision boundary for the selector?  If you assume a simple function class -- e.g. linearly separable in low-dimensions I assume it's easy to identify -- but generally the selector region may itself has a complex boundary, which may require regularization to learn.  How do you balance the regularization of the selector vs. the selector accuracy vs. the predictor accuracy (vs predictor regularization)?  It seems practically daunting.

5) You mention that the predictable region has to be high-signal-to-noise. How sensitive it is to this? For example if there are two regions, one already noisy, and another completely unpredictable, how quickly does it break down with the SNR?


**Summary Of The Paper:**

This paper considers supervised learning with abstention -- where the learning method can decide to make prediction in some region of the feature space, and declare the rest of the feature space unpredictable. The setting is related to but different from the classical selective prediction, and to prediction uncertainty quantification (i.e. imbuing predictions with calibrated confidence intervals). Classical selective prediction balances coverage vs. accuracy -- whereas in the setting here the authors assume that there is a "true" split into an informative and uninformative partition of the feature-space,  and the goal is to both recover the partition and learn a good model on the informative partition.  The paper focuses on theoretical analysis, and also has experimental results suggesting that if the data is generated according to the author's model, than a heuristic algorithm inspired by the analysis indeed outperforms other selective prediction baselines.

**Summary Of The Review:**

This paper proposes a novel formulation supervised learning with abstention. It assumes that feature space can be split into an informative region, and a completely unpredictable region.  It offers detailed theoretical analysis showing that recovery is possible, and a heuristic practical algorithms to both recover the partition of the feature-space, as well as learning the model over the predictable region.  The setting is interesting -- but I suspect it's very artificial, so I'd be very curious to know to what extent this has relevance to applications which have no clear-cut partitions (or to traditional selective inference).

---

> ### Author Response · Authors · 2021-11-19
> **Response to Reviewer QnTY - Cont'd**
>
> ***Q4. How do you incorporate the complexity of the decision boundary for the selector? ... How do you balance the regularization of the selector vs. the selector accuracy vs. the predictor accuracy (vs predictor regularization)? It seems practically daunting.***
>
> ___Answer___:  In theory, the recoverability of the selector depends on the complexity of the decision boundary and when the decision boundary is complex we will need more data to reduce the statistical error.
>
> In practice, we count on the capacity of neural networks (NN) to approximate the complex decision boundary, and hope the NN output is not too `complex’ due to implicit regularization techniques such as weight decay, batch normalization, SGD. This is similar to many existing works investigating the generalizability of deep neural networks.
>
>
> ***Q5. You mention that the predictable region has to be high-signal-to-noise. How sensitive is it to this? For example if there are two regions, one already noisy, and another completely unpredictable, how quickly does it break down with the SNR?***
>
> ___Answer___:  Empirically our method is robust w.r.t this relaxation and we believe our theoretical analysis can also be extended accordingly. Here is a new empirical study w.r.t such relaxation (we only studied the challenging case where 20% of the data is informative). We allow the label of informative data to be flipped uniformly with a constant probability so that it has label noise, relaxing the assumption that the informative data can be perfectly classified. It can be observed that the performance deteriorates slowly as the label noise level increases. When the noise level reaches 60%, the method is broken. With higher noise levels, the selector is forced to move the noisy data to the uninformative data category. The classifier gets less data and is harder to train.
>
> ___Table G.___ MNIST - Unknown $\alpha$  - With Label Noise - 60000 Uninformative / 15000 Informative
>
> |   label_noise | SR(\%) | precision | recall |
> |---:|:---|:---|:---|
> | 0% | 11.29$\pm$1.67  | 1.00$\pm$0.00   | 0.74$\pm$0.07 |
> | 20% | 7.68$\pm$0.00   | 1.00$\pm$0.00   | 0.59$\pm$0.00 |
> | 40% | 13.04$\pm$3.27  | 1.00$\pm$0.00   | 0.52$\pm$0.03 |
> | 60% | 54.19$\pm$39.66 | 0.63$\pm$0.41   | 0.32$\pm$0.40 |

---

> > ### Comment · Reviewer_QnTY · 2021-11-28
> > **Thank you for the feeback.**
> >
> > I'd like to thank the authors for a detailed and comprehensive feedback, this was very helpful.  The hypothesis of 'certain random pockets/scenarios of predictability' is curious, but I still am not sure what evidence is there to support it, and whether the decision boundary for detecting such rare pockets isn't overly complex (and e.g. it may be impacted by extreme class imbalance). I'm also not convinced that using standard DNN regularization techniques will be obvious when you have to use it at the same time for both the selector and the predictor in this setting.  The failure of uncertainty quantification in your simulation is surprising, it does seem that the method may have good use cases and I encourage the authors to find a convincing (realistic) applied example. I do find the paper interesting -- but I keep my current rating.

---

> > > ### Author Response · Authors · 2021-11-28
> > > **Thank you for your further feedback**
> > >
> > > We thank you for reading our response and providing further feedback. We would like to clarify  just in case there is a misunderstanding:
> > >
> > > ***Q6. I'm also not convinced that using standard DNN regularization techniques will be obvious when you have to use it at the same time for both the selector and the predictor in this setting.***
> > >
> > > __Answer:__
> > >
> > > The DNN models for classifier and for selector are not the same model. We used two individual models. Each has its own training loss and regularization. Thus even if the informative/uninformative boundary is different from the classification boundary, the selector and classifier can still fit them well.
> > >
> > > Our empirical evaluation of SVHN also studies the case where the selector decision boundary is complex (arguably as complex as classification task). Even though the scenario where data is very limited (2k out of 11k is informative), the method can still manage to achieve reasonable performance.
> > >
> > > Please feel free to let us know if there are further questions. Thanks again for your constructive feedback and your  time.

---

> > > > ### Comment · Reviewer_QnTY · 2021-11-29
> > > > **thanks!**
> > > >
> > > > Thanks, yes, I'm on board that the two models are separate and their regularization parameters are not tied.  All I meant to say is that I would expect choosing the regularization parameters jointly to be significantly harder than for a standard standalone prediction task.

---

> > > > > ### Author Response · Authors · 2021-11-30
> > > > > **Thanks for the further reply**
> > > > >
> > > > > Thanks for the clarification. We agree that the problem itself becomes more challenging when one needs to recover the selector and classifier jointly as it requires more samples (which depends on the complexity of the selector). We look forward to exploring settings with more complex selector decision boundaries to see if the advantage of our algorithm compared to baselines still holds.

---

> ### Author Response · Authors · 2021-11-19
> **Response to Reviewer QnTY**
>
> Thank you for your constructive feedback. We address your questions/concerns as follows:
>
> ***Q1. To me the model seems entirely artificial. Showing how this translates to a more realistic example would be helpful.***
>
> ___Answer___: Our proposal for the noisy data generation setting is originally motivated by problems in financial time series. While most of the time stock returns are nearly purely random, they become predictable in certain rare pockets/scenarios. Examples of these scenarios include when a large market order arrives, when good/bad news are disseminated, or under certain corporate actions, etc. Identifying these small windows of predictability among random data is exactly the mission of a financial machine learning algorithm. This inspired our thinking.
>
> Our assumption of a completely separable noisy/clean data support is close to but not exactly capturing the real-world scenario. The random data can have a small amount of signal, and the predictable data can have a small amount of uncertainty. There can also be a small region of phase transition between the two (not a smooth transition). Extending our work to these settings is not trivial and will definitely be our next step. The analysis, of course, will require more advanced tools for real-valued functions e.g. Rademarcher complexity.
>
> ***Q2. I am surprised that using methods which allow uncertainty quantification -- e.g. maybe bootstrap resampling, or say Gaussian processes would have a hard time in this setting. …  (e.g. there's a nice paper "On Calibration of Modern Neural Networks" from Killian Weinberger's group, and also well known solutions, for example from Charler Elkan and Bianca Zadrozny, "Obtaining calibrated probability estimates for classifiers".***
>
> ___Answer___: Indeed, our problem can potentially be tackled by the uncertainty quantification methods.  We have compared with the suggested network calibration method (Guo et al. 2017) and provide its performance in Table-F. We use the same setting as we do for all other baselines. We select the top estimated $\alpha$ fraction of data based on the calibrated confidence. The confidence calibration is based on the test set, which is in favor of this baseline.
>
> We observe that the calibration method performs similarly to selective learning methods and is not very effective. The major reason is that the calibration is a post processing step which does not provide any regularization in the training stage to help improve the classifier. It cannot prevent the classifier from overfitting to the uninformative data. In the future, it is possible to adapt the method to our setting through a tighter coupling of the confidence calibration and the classifier training.
>
> In fact, we believe confidence calibration is a harder problem than selecting uninformative data; the former learns a real-valued function and the latter learns a boolean function. This perspective can potentially help relax our assumption of the separation of uninformative / informative data support.
>
> ___Table F.___ MNIST - Unknown $\alpha$  - Calibration
>
> | uninform_data/inform_data   | SR(%)   | recall   | precision   |
> |:----------------------------|:--------------------|:-------------------|:----------------------|
> | 60000/60000                 | 18.75$\pm$1.25      | 0.96$\pm$0.02      | 0.92$\pm$0.02         |
> | 60000/45000                 | 27.32$\pm$2.53      | 0.92$\pm$0.04      | 0.83$\pm$0.03         |
> | 60000/30000                 | 39.64$\pm$3.03      | 0.89$\pm$0.04      | 0.74$\pm$0.04         |
> | 60000/15000                 | 50.07$\pm$1.89      | 0.88$\pm$0.02      | 0.58$\pm$0.02         |
> | 15000/15000                 | 44.47$\pm$1.06      | 0.91$\pm$0.01      | 0.63$\pm$0.01         |
> | 15000/11250                 | 53.71$\pm$1.21      | 0.86$\pm$0.04      | 0.52$\pm$0.03         |
> | 15000/7500                  | 64.61$\pm$0.81      | 0.83$\pm$0.03      | 0.40$\pm$0.01          |
> | 15000/3750                  | 77.16$\pm$0.92      | 0.83$\pm$0.03      | 0.26$\pm$0.01         |
>
> ***Q3. If instead of trying to identify the "predictable region" -- you try to apply / modify the method to attack the original selective prediction -- is it competitive with existing approaches? My suspicion is that the approach is quite dependent on having data with a well-separated true informative/uninformative partition -- and going after support recovery.***
>
> We do not expect it will be easy to adapt our algorithm to selective learning. Our method, especially the selective loss, is designed to recover the indicator function of informative/uninformative data. We are approximating a ground truth selector point-wise (without direct supervision). Whereas, in selective learning, no data is better than others. One is trying to achieve a given global criterion for the selector, e.g., coverage.

---

### Official Review · Reviewer_vqyY · 2021-11-04

**Correctness:** 4
**Technical Novelty And Significance:** 3
**Empirical Novelty And Significance:** Not applicable
**Recommendation:** 6
**Confidence:** 3

**Main Review:**

Strengths.

The paper is organized well with clear problem formulation, definitions & assumptions. The methodology is supported with theoretical results and implemented by an iterative algorithm. While the problem formulation requires uninformative samples are separated from informative ones, the authors also tested more challenging settings in the experiments.

Weaknesses.

There are previous papers related that are missing here. For example, "Combating Label Noise in Deep Learning Using Abstention" by Thulasidasan et al. also seems to do very similar things and has similar title, but is not discussed in the related work. Also, some descriptions to previous works are not accurate, e.g., "While above works assume the presence of noisy data only in the training stage" -- some of the works above are in the parameter recovery setting, so it is slightly unfair to compare without the problem setting described clearly.




**Summary Of The Paper:**

The paper considers a setting where the label is random noise if the input is in an uninformative subspace, provides theoretical analysis for the estimated predictor/selector that minimizes classification risk/a novel selector loss, proposes a practical iterative algorithm to approximate the estimators based on MWU, and tests on semi-synthetic experiments.

**Summary Of The Review:**

The paper is well written in general. The descriptions on related literature need some improvement.

---

> ### Author Response · Authors · 2021-11-19
> **Response to Reviewer vqyY**
>
> Thank you for your constructive feedback. They are very helpful in improving the quality of the paper. Below we address your concerns:
>
> ***Q1. "Combating Label Noise in Deep Learning Using Abstention" by Thulasidasan et al. also seems to do very similar things and has similar title, but is not discussed in the related work.***
>
> __Answer__: Thank you for suggesting this interesting work. The method in (Thulasidasan et al. 2019), called DAC, is proposed to tackle a different task- noisy labels. The major difference is that because we aim to learn a selector that can abstain from uninformative data in both the training and inference stage, whereas DAC needs to learn a good classifier with noisy dataset. Indeed, DAC can be used to deal with our problem after non-trivial adaptation. It is actually similar in philosophy to one of our selective learning baselines, DeepGambler; both use extra neurons to indicate unsureness and to make abstention. We will cite and discuss in the final version.
>
> ***Q2. Also, some descriptions to previous works are not accurate, e.g., "While the above works assume the presence of noisy data only in the training stage" -- some of the works above are in the parameter recovery setting, so it is slightly unfair to compare without the problem setting described clearly.***
>
> __Answer__: Agreed. We will revise in the paper that existing label noise works are focusing on recovering the true classifier despite noisy observations. This is a different problem from ours.

---

### Official Review · Reviewer_8Gu4 · 2021-11-08

**Correctness:** 4
**Technical Novelty And Significance:** 3
**Empirical Novelty And Significance:** 2
**Recommendation:** 3
**Confidence:** 4

**Main Review:**

Strength & Weakness: I am conflicted about the main structure of the paper of cleanly separated completely noisy and completely clean data with both aspects realisable. In my opinion this situation is quite unrealistic, which meaningfully impairs the study. Nevertheless, this seems to be the analogue of realisability relevant to selective classification, and thus a natural setting to investigate tight bounds on the basic aspects of the problem. That said, I don't think the paper makes this case sufficiently well, instead posing the setting as a sort of Huber-type contamination of the data that persists to test time, but is kind enough to be entirely separate from the real data - this is quite unconvincing to me.

Strengths: Modulo the above, I find the basic idea of the scheme and the loss embodying it to be quite interesting, and the analysis supporting it is straightforward, and natural. I find the heuristic to be even more interesting, in particular due to the multiplicative weighting idea, which I think is an elegant way of reinforcing the attention of $f$. The empirical results are, of course, stark and impressive, but I have some reservations about the same (see below)

Weakness:

- Theoretically, there are two gaps - the major one, in my opinion, is that the analysis requires a priori knowledge of $\alpha$ to appropriately set $\theta$ for the loss. This I think weakens the results significantly. One way to ameliorate this would be to extend the results so that an upper bound on $\alpha$ is sufficient (which is a much more realistic assumption, although of course ideally the method should adapt to $\alpha$). Also important, but less so, is the lack of a lower bound for the problem (it is clear that reliably estimating $f^*$ would necessitate the use of $\Omega( (\alpha \epsilon)^{-2})$ samples, I don't think this fact is all that interesting).

- Specifically regarding the experiments, I have questions about the baselines, and the proposed method's hyperparameter selection.

    - Lack of explanation of the results: The empirical results presented are starkly positive, but the exposition offered does not clarify what aspect of the scheme is buying this performance . For instance, all the methods seem to estimate the underlying $g^*$ well, but the competing methods cannot offer appropriate labels. Is this because of the repeated multiplicative re-weighting in the loss for $f$? Do the results hold if this is turned off? Does a similar iterative training improve the risk for the competing methods? I think ablative analysis along these lines is crucial for understanding the contribution of this paper.

    - Lack of clarity about baselines.
        - With empirical performances so starkly different, I cannot but help question if the baselines are being fairly executed. The big issue here is whether the hyperparameters for these methods are being selected in an appropriate way - this is bolstered by the situation of Tables 5-8, where $\alpha$ is assumed to be available to the algorithms, and the baseline hyperparameters are thus  presumably being set appropriately, and the performance gaps are much reduced. This leads me to the following questions:
            - Does this procedure given on page 8 yield a decent estimate of $\alpha$? How does performance vary if you train for more epochs?
            - In the appendix, it is stated that default hyperparameters are used for each method. What does this mean for something like selective net?
            - The baseline 'Confidence' is never described. Is this the method proposed by Geifman & El-Yaniv 2017? Something else? Does this method have hyperparameters? How are they tuned?
 I understand that choosing the hyperparameter in these prior methods for this particular scenario may be considered beyond the scope of this paper, but in my opinion unless it is justified that the hyperparameters are being selected in a meaningful way, this presentation is meaningless.

    - Hyperparameter selection for the proposed method: how precisely is $\beta$ selected? This is never described in the text, and the only reference to it appears in Appendix D where it is said that the default parameter $\frac{\theta (1-\alpha)}{\alpha (1-\theta)}$ is used. This is both non-informative, because the question then becomes how is $\theta$ selected, and troubling, because it appears that $\alpha$ is being used to pick $\beta$. Is this the case?


- Relation to prior literature:

I think the selective classification literature is slightly mischaracterised, in the sense that it is not restricted to coverage oriented designs, as is suggested on page 3. Three explicit references theoretically and methodologically studying low-error selective classification are [1,2,3], all of which give PAC-style definitions regarding attaining low error selective classifiers, and [1] shows various computational reductions regarding the learnability of selective classifiers, and [2,3] provide sample complexity guarantees. In fact I believe Proposition 4 in [3] is directly applicable to the situation of this paper. Further, [3] also proposes a method that empirically dominates both the selective net and deep gamblers methods, and so might be good to compare to.

Additionally, I think [4] might be a pertinent reference, since it too proposes an alternating minimisation strategy for learning selective classifiers (although I think there's plenty of novelty in the presently proposed method beyond this).

[1] Adam Tauman Kalai, Varun Kanade, and Yishay Mansour. Reliable agnostic learning. Journal of Computer and System Sciences, 2012.

[2] Durmus Alp Emre Acar, Aditya Gangrade, and Venkatesh Saligrama. Budget Learning via Bracketing. AISTATS, 2020.

[3] Aditya Gangrade, Anil Kag, and Venkatesh Saligrama. Selective Classification via One-Sided Prediction. AISTATS, 2021.

[4] Feng Nan, and Venkatesh Saligrama. Adaptive Classification for Prediction Under a Budget. NIPS, 2017.

---

Questions and Comments:

- Assuming that the experiments are kosher, do you have hypotheses for how the advantage is being gained over the baselines? I noticed that the precision and recall (presumably of the selection with respect to a noisy/not-noisy oracle) of all methods are similar. Does this mean that the other methods are simply not learning the correct predictions on the domain they do select?

- For the theoretical parts, I haven't checked this thoroughly, but it seems like it should be possible to get away with using about $\frac{d(\mathcal{F}) + d(\mathcal{H})}{\min(\alpha, 1-\alpha)\epsilon}$ labeled and $\frac{d(\mathcal{H})}{\alpha^2 \epsilon^2}$ labeled datapoints (up to $\log$s) by picking $g$ to be the maximiser of ${\hat{P}}_\{\lnot \ell\}(g = 1)$  subject to the constraint that $\hat{P}_\{ \ell\}(f \neq Y, g = 1) = 0$, where $\hat{P}$s are unlabeled and labelled empirical measures. Note that this is just the scheme of [3]..


Quibbles:

- I don't think $f^*$ as defined really makes sense. It is ostensibly defined on all of the space, but it clearly cannot be uniquely defined here in even mildly rich settings - for instance, if $\mathcal{F}$ has functions that agree on $\Omega_I$ but differ on $\Omega_D$.
- In section 3, the abstention criterion suddenly switches from $g = -1$ to $g < 0$. While mathematically identical, such devices are commonly used in the ML literature to semantically adjust  functions from 'hard' to 'soft'. Is this the intention here? If not, perhaps the authors can stick to one presentation? More importantly, in section 3.3 $g$ suddenly switches to being $[0,1]$ valued, and the abstention criterion to $g < 0.5.$ I understand that this is for the sake of the cross-entropy loss, but this is needlessly confusing - I think consistency here saves your readers needless recalibration.
- It is strange that $g$ is taken to lie in $\mathcal{H}$ rather than $\mathcal{G}.$


**Summary Of The Paper:**

The paper is concerned with selective classification in a stylised 'realisably noisy' data model, wherein the support of the input distribution is partitioned into two chunks, the "informative" $\Omega_I$ and the "uninformative" $\Omega_U,$ such that
- The labels are completely noisy ($\mathrm{Bern}(1/2)$) if the input lies in $\Omega_U$.
- The labels are completely clean if the input lies in $\Omega_I$.
- The learner has access to hypothesis classes $\mathcal{F}, \mathcal{H}$ such that
    - There exists an $f^* \in \mathcal{F}$ with $f^*(x) = y$ on $\Omega_I$.
    - There exists a $g^* \in \mathcal{H}$ with $g^*(x) = 2\mathbf{1}\{x \in \Omega_I\} - 1.$

The paper studies the problem of learning a pair $(f,g),$ with $g$ serving as a selector, and $f$ as a predictor, such that $g \approx g^*$ and $f \approx y$ on $\{g \ge 0\}$ (and $\approx f^*$ on $\Omega_I$ if $g$ is learnt well), with the concrete goal of attaining both small selective risk ($P(f^*(x) \neq y|g(x) \ge 0)$) and both low false alarm and low missed detection for $g$ with respect to $g^*$. As is standard in selective classification, supervision of the value of $g^*$ is not given, and instead only $(x,y)$ pairs are provided.

This is approached by designing a new loss for learning a selector given a predictor $f$, denoted $W(g;f, \theta)$, that basically uses $\mathbf{1}\{f(x) = y\}$ as proxy supervision for when $g < 0$ or $g > 0$ is preferred, but weighted by a factor to account for the prevalence of uninformative versus informative data. It is first shown, using uniform convergence techniques, that if the indicator ERM problem can be solved, then given  if $f^*_{S_n}$ is taken to be a minimiser of standard empirical risk, and $g^*_{S_n}$ to be the minimiser of an empirical version of $W(g;f^*_{S_n}, \theta)$ for appropriately chosen $\theta$ (in a way which depends on the prevalence of informative data $\alpha$), then these goals can be attained in a PAC sense, with sample complexity scaling as $\tilde\{O\}( \frac{d_\{\mathrm{vc}\}(\mathcal{F}) + d_\{\mathrm{vc}\}(\mathcal{H}) }{\epsilon^2 \alpha^2})$.

The paper then switches gears, and a alternating minimisation based heuristic method for practically learning $(f,g)$ is proposed. The idea is to first learn (soft functions) $f$ and then $g$ using relaxed versions of the above losses, and then multiplicatively increase the weight of points that are selected by the $g$ in their contribution to the loss for $f$, and repeat. The remainder of the paper is devoted to empirical evaluation of this method in two situations - firstly, uniformative (randomised labels) MNIST data is given along with varying amounts of clean Fashion MNIST data; and secondly, part of the classes from the SVHN dataset have their labels randomised to represent uninformative data, while the rest remain clean. While the scheme is observed to have similar performance in how well the selector identifies the underlying $g^*$, the proposed method has remarkably better selective risk compared to baselines from the recent selective classification literature.

**Summary Of The Review:**

On the whole, I think that while I don't find the scenario posed by the paper convincing, I do find the concrete setting itself to be so. The scheme proposed for the same is interesting, and, taken at face value, very effective. Nevertheless, I right now am not convinced that the baselines are being compared to correctly, which needs to be justified, and further, I think that ablative experiments illuminating how the aspects of the proposed contribute to performance are needed. I think without these, though, the theoretical study ultimately produces a sample complexity analysis for a method that uses the knowledge of a key underlying parameter, and even then is not shown to be tight, which is a bit too weak, and I think needs more work.

With the above consideration, I have reservations about recommending acceptance as the paper stands. I am willing to budge on this if the experiments are clarified, or if the analysis can be improved.

---

> ### Author Response · Authors · 2021-11-19
> **Response to Reviewer 8Gu4 - Cont'd**
>
> ***Q12. In the appendix, it is stated that default hyperparameters are used for each method. What does this mean for something like a selective net?***
>
> ___Answer:___ For experiments presented in Table 1$\sim$4 in the paper where there is no complete informative validation dataset, the choice of hyper-parameter cannot be determined by an HPO procedure under different levels of $\alpha$. Thus the default parameter, which is the parameter recommended in the original paper i.e weight for coverage rate penalty $\lambda=32$ and weight for selector loss $a=0.5$. For DeepGambler, the sole hyper-parameter “reward” should lie between 1~10 and we choose the default value recommended by the author which is 2.2.
>
> We provide an additional ablation study about the key hyper-parameters of baselines (Figures 5 and 6 in the Appendix of the updated manuscript). We observe that the baselines ___consistently perform poorly___ within a large range of values of their key hyper-parameters.
>
> ***Q13. The baseline 'Confidence' is never described. Is this the method proposed by Geifman & El-Yaniv 2017? Something else? Does this method have hyperparameters? How are they tuned? I understand that choosing the hyperparameter in these prior methods for this particular scenario may be considered beyond the scope of this paper, but in my opinion unless it is justified that the hyperparameters are being selected in a meaningful way, this presentation is meaningless.***
>
> ___Answer:___ Thank you for pointing this out, as we should have been more explicit about this baseline. For baseline confidence, we select data points that have top $\alpha$ confidence. For experiments where $\alpha$ is not known, we use estimated $\hat{\alpha}$ here.
>
> Note that the method proposed by Geifman & EI-Yaniv 2017 comes with a totally different intuition, which tries to search for an optimal confidence threshold such that when selecting according to this threshold, the true risk will be upper bounded by the found quantity.  However, in our setting, this risk requirement is irrelevant, since we aim to recall as much informative data as possible and achieve low risk conditioning on the selection result. The final selective risk will be mainly determined by the fraction of informative data and is unlikely to be revealed to the user in a practical setting.
>
> ***Q14. Hyperparameter selection for the proposed method: how precisely is $\beta$ selected? it appears that $\alpha$ is being used to pick $\beta$. Is this the case?***
>
> ___Answer:___ We do not need the exact value of $\alpha$ to choose $\beta$. As stated earlier in Question 1 Part III, $\beta$ only depends on the upper bound of $\alpha$. Also refer to Theorem 5 in the Appendix of the revision.
>
> Empirically, when $\alpha$ is unknown, we used a fixed $\beta$ in all experiments. We provide a new ablation study about the robustness of the choice of $\beta$. As shown in Figure 4 in the Appendix of the revision, the performance is consistently strong regardless of the choice of $\beta$.
>
> ***Q15. Further, [3] also proposes a method that empirically dominates both the selective net and deep gamblers methods, and so might be good to compare to.***
>
> ___Answer:___ We would love to compare with [3]. However, we couldn’t find its source code, and didn’t have the time to implement the method from scratch.

---

> ### Author Response · Authors · 2021-11-19
> **Response to Reviewer 8Gu4 - Cont'd**
>
> ### ___PART III- Response to concerns on the  Theory:___
>
> ***Q4. Theoretically, there are two gaps - the major one, in my opinion, is that the analysis requires a priori knowledge of $\alpha$ to appropriately set $\theta$ for the loss. This I think weakens the results significantly. One way to ameliorate this would be to extend the results so that an upper bound on $\alpha$ is sufficient.***
>
> ___Answer:___ This is a very good point! Indeed, it is not hard to rewrite our theorem so that it only depends on an upper-bound of $\alpha$ (i.e., $\alpha$ is bounded away from 1). We provide the revised theorem (Theorem 5 in Appendix) in the updated manuscript.
>
> To be precise, we rewrite our selector loss as $Loss(g; f^*_{S_n}, S_n, \beta) = \sum_{i=1}^{n}\beta \mathbf{1}\left[f^*_{S_n}(x_i) \neq y_i, g(x_i) >0\right] + \mathbf{1}\left[f^*_{S_n}(x_i)=y_i, g(x_i) \leq 0 \right]$ and now our method depends on a single hyperparameter $\beta$. This is indeed how we implemented our algorithm. If one knows $\alpha \leq 1-\xi$, to achieve a PAC guarantee similar to the main theorem, it suffices to pick $\beta = \frac{\theta}{\alpha}\frac{1-\alpha}{1-\theta} \in (1+\frac{1}{73\xi-1}, min(1+\frac{\xi}{2(1-\xi)},10))$ (which is a wide range when $\xi$ is a reasonable constant). In our response regarding experiments, we will demonstrate that our result is very robust with regard to the choice of $\beta$.
>
>
> ***Q5. Also important, but less so, is the lack of a lower bound for the problem (it is clear that reliably estimating $f^{\ast}$ would necessitate the use of $\Omega( \frac{1}{\alpha^2 \varepsilon^{2}}) $samples, I don't think this fact is all that interesting).***
>
> ___Answer:___ We assume the reviewer meant that we do not have an information theoretical lower bound, and the provided ERM lower bound seems trivial.
>
> We strongly believe that our ERM lower bound is nontrivial, ERM being arguably the most important estimator in ML.
> Furthermore, our construction of the lower bound is technically challenging. It requires a hard case for our noisy generative process, where both the separability assumption and realizable assumption for informative data are satisfied. The ground truth selector also needs to have VC-dimension comparable to the classifier hypothesis class.
>
> ***Q6. It seems like it should be possible to get away with using about  $\frac{d_{vc}( \mathcal{F})+d_{vc}(\mathcal{H})}{min(\alpha,1−\alpha)\epsilon}$ labeled and $\frac{d_{vc}(\mathcal{H})}{\alpha^2\epsilon^2}$ labeled data points (up to logs) by picking $g$ to be the maximiser of  $P_{\neg \ell}(g=1)$ subject to the constraint that  $P_{\ell}(f\neq Y,g=1)=0$, where $P_S$ are unlabeled and labelled empirical measures. Note that this is just the scheme of [3].***
>
> ___Answer:___ We assume that the reviewer meant  “$\frac{d_{vc}(\mathcal{F})}{\alpha^2\varepsilon ^2}$”. Otherwise, it is impossible as learning $f^{\ast}$ will need $\Omega(\frac{1}{\alpha^2\varepsilon^2})$ samples.
>
> If he indeed mean “$\frac{d_{vc}(\mathcal{F})}{\alpha^2\varepsilon^2}$”, it is unclear whether such a solution is possible. First, the sample complexity in [3] is $\frac{d_{vc}(\mathcal{H})}{\varepsilon^2}$. It is not trivial, if at all possible, to prove $\frac{d_{vc}(\mathcal{F})+d_{vc}(\mathcal{H})}{min(\alpha,1−\alpha)\varepsilon} $ sample complexity for the algorithm in [3].
>
> Second, the proposed scheme from [3] requires an explicit split of the training set into two subsets, for training the classifier and selector separately. The split requires knowing the complexity of classifier and selector, which is impossible in our setting.
> Finally, enforcing $P_\ell (f \neq Y,g=1)=0$ will incur more false negative error since empirically evaluated f can not achieve 0 error. The statistical error can be rather large when data is limited. Consequently, the risk level of $f$ can be far from 0.
>
> ***Q7. I don't think $f^{\ast}$ as defined really makes sense. It is ostensibly defined on all of the space, but it clearly cannot be uniquely defined here in even mildly rich settings - for instance, if $\mathcal{F}$ has functions that agree on $\Omega_I$ but differ on $ \Omega_U$.***
>
> ___Answer:___ We allow  $f^{\ast}$ to be defined over the whole domain for ease of exposition. It is true that there can be different  $f^{\ast}$ that are the same on $\Omega_I$ but different on $\Omega_U$. We may choose any of them as our true $f^{\ast}$ without affecting our result; on $\Omega_U$,  $f^{\ast}$ is not used to sample labels, and thus is not involved in the selective loss or the classification loss. We will clarify this in the final version of the paper.
>
> ***Q8. Consistency about $g$ [in the presentation]***
>
> ___Answer:___ Thank you for the suggestion. Will change in the revision.
>
> ***Q9. It is strange that g is taken to lie in $\mathcal{H}$ rather than $\mathcal{G}$.***
>
> ___Answer:___ Thank you for pointing this out. We will improve the presentation in the final version.

---

> ### Author Response · Authors · 2021-11-19
> **Response to Reviewer 8Gu4 - Cont'd**
>
> ### ___PART II-Response to major concerns:___
>
> ***Q1.  In fact I believe Proposition 4 in [3] is directly applicable to the situation of this paper.***
>
> ___Answer:___ The algorithmic scheme and Proposition 4 in [3] cannot be applied here. Method in [3] is aimed at classical selective learning problems: it searches for a selector that maximizes the coverage while maintaining a risk level.
>
> Proposition 4 gives an existence PAC learnable guarantee based on a conceptual algorithm requiring an oracle that provides all possible classifiers below a given risk level. It is unclear how to approximate such an oracle and translate it into a practical solution for our problem. It is also unclear how to apply the proposition [4] into our theoretical analysis; our analysis is based on a practical empirical risk minimization oracle.
>
> It is also challenging to apply the algorithmic scheme in [3] to our problem. While the algorithmic scheme in [3] does not require knowing the fraction of the informative data, it does require something else. The method maintains a risk level for the classifier, so it requires knowing the prediction quality of the classifier as input. In our problem, since we do not know which samples are informative or uninformative, it is very difficult to reliably estimate the risk level of an empirically learned classifier.
>
>
> ***Q2. I am conflicted about the main structure of the paper of cleanly separated completely noisy and completely clean data with both aspects realizable. In my opinion this situation is quite unrealistic, which meaningfully impairs the study.***
>
> ___Answer:___ Our proposal for the noisy data generation setting is originally motivated by problems in financial time series. While most of the time stock returns are nearly purely random, they become predictable in certain rare pockets/scenarios. Examples of these scenarios include when a large market order arrives, when good/bad news are disseminated, or under certain corporate actions, etc. Identifying these small windows of predictability among random data is exactly the mission of a financial machine learning algorithm. This inspired our thinking.
>
> Our work is the first, to our knowledge, to formulate the noisy data generation process and to study learning in this setting. We believe our method to be nontrivial, and our analysis is technically involved. We showed strong empirical results with extensive experiments.
>
> Our assumption of a completely separable noisy/clean data support is close to but not exactly capturing the real-world scenario. The random data can have a small amount of signal, and the predictable data can have a small amount of uncertainty. There can also be a small region of phase transition between the two (not a smooth transition). Extending our work to these settings is not trivial and will definitely be our next step. The analysis, of course, will require more advanced tools for real-valued functions e.g. Rademarcher complexity.
>
> ***Q3. Lack of explanation of the results. Results are starkly positive. Do you have hypotheses for how the advantage is being gained over the baselines? I think ablative analysis along these lines is crucial for understanding the contribution of this paper.***
>
> ___Answer:___ The intuition is as follows: __when $\alpha$ is not given__, existing methods are not designed to deal with such scenarios, and thus perform poorly. These methods need $\alpha$ as an input. However, the estimation of $\alpha$ is a challenging task. With a poorly estimated $\alpha$, these methods will have poor performance (we will provide those ablation studies later in the response). Our method does not have such an issue, since the hyperparameter $\beta$ only depends on an __upper bound on $\alpha$__. Indeed, in all the experiments where alpha is unknown, we use a fixed beta because it is robust w.r.t $\alpha$. We provide an ablation study to showcase this stability in Appendix F. To summarize, we observe that different values of beta all behave reasonably well and none lead to dramatic performance deterioration.
>
>
>
> __When $\alpha$ is known__, our method outperforms baselines marginally. The advantage is due to the multiplicative weights update (MWU), which allows our approach to concentrate its entire learning capacity on just the informative data. We also provide an ablation study of this mechanism later in the response.
>
> We also note that __our empirical evaluations on the baselines are fair and thorough__. We did perform the full rigorous hyperparameter optimization (HPO) for all methods, and the HPO results  will be released together with the source code.

---

> ### Author Response · Authors · 2021-11-19
> **Response to Reviewer 8Gu4**
>
> ### __PART I - Summary__
>
> Thank you for the feedback. All comments have been extremely helpful. We summarize our response here and aim to address them below (and of course incorporate them in our final version) in the following 3 broad categories:
>
> __Problem is unrealistic, motivation is unclear__: we provide motivation for our setup and argue that it is firmly rooted in real-world considerations. We also position our proposed algorithm relative to current benchmarks and provide the intuition for why our approach succeeds where others fail.
>
> __Relaxing assumption in the theory, PAC guarantee without knowing exact $\alpha$__: thanks to the reviewer’s excellent suggestion, we have added a new Theorem in Section E that achieves similar PAC guarantees as the main theorem, but only requires an upper bound on the value of $\alpha$. This further clarifies why our results appear stable under a wide range of values of our main hyperparameter, $\beta$.
>
> __Experiments need clarification and more ablation studies:__
>
> We provide an intuitive explanation backed by empirical results for why our method outperforms baselines when $\alpha$ is known (Tables 5-8 in the paper). Running our algorithm without the multiplicative weights update mechanism leads to performance drop - MWU is the reason why our approach slightly beats the benchmarks when $\alpha$ is known (Table A below)
>
> We provide empirical results showing why we have such a big gap compared to the baseline when $\alpha$ is unknown. We train for longer when estimating fraction of informative data - this goes towards demonstrating that it’s difficult to estimate the percentage of informative data without proxies (such as the classifier’s performance in our case), especially when you have access to relatively little data (Table B ~ E below)
>
> We provide empirical results showing that our method is robust w.r.t key hyperparameter $\beta$ (Figure 4 in the Appendix). We observe that the baselines consistently perform poorly within a large range of values of their key hyper-parameters (Figures 5,6 in the Appendix.).
>
> We have also provided more background on previously-run experiments and clarified the setup of the Confidence benchmark that was unfortunately left off the original draft
>
> We will add the suggested references [1,2,3,4] in the revision and provide comparisons. We find  [3] to be particularly interesting; however, because it studies a different problem, we don’t believe that its schema can be applied directly here (will address this in detail later on).

---

> ### Author Response · Authors · 2021-11-26
> **Soliciting Post-Rebuttal Response**
>
> Dear Reviewer 8Gu4,
>
> We understand you must be busy during the Thanks Giving period and we are grateful for your time. We are wondering whether our initial responses have addressed your concerns. Specifically, We have
> 1) provided an example justifying our model assumptions in real world scenarios;
> 2) extended our theorems with assumptions you suggested;
> 3) clarified the difference between our work and the existing works you provided, and why existing methods do not apply to our problem easily;
> 4) clarified experimental results with more insights and ablation studies.
>
> We ask you to kindly let us know if there is any other confusion, which possibly may prevent you from assessing the merits of our method.
> We would be more than happy to clarify any such doubts during the discussion period.
>
> Thank you again for your time, and happy Thanksgiving!
>
> Sincerely,
> Authors

---

### Author Response · Authors · 2021-11-19
**Overall Response.**

We thank all reviewers for the valuable time and effort they spend reading and reviewing our paper. We appreciate your constructive suggestions and will improve the paper accordingly. Below we address specific concerns one-by-one.

---

### Decision · Program_Chairs · 2022-01-20

**Decision:**

Reject

**Comment:**

This paper studies a learning scenario in which there exist 2 classes of examples: "predictable" and "noise". Learning theory is provided for this setting and a novel algorithm is devised that identifies predictable examples and makes predictions at the same time. A more practical algorithm is devised as well. Results are supported by experiments.

Reviewers have raised a number of concerns (ranging from how realistic this settings is to missing references). Overall they found this work interesting and relevant to ML community and appreciate the effort that authors have put in in their thoughtful response. However, after a thorough deliberation conference program committee decided that the paper is not sufficiently strong in its current form to be accepted.